# Chromatin accessibility landscape and regulatory network of high-altitude hypoxia adaptation

Jingxue Xin[1,2,3,4,5,12], Hui Zhang[1,3,12], Yaoxi He [1,3,5,12], Zhana Duren[6,11,12], Caijuan Bai[7,12], Lang Chen[2,5], Xin Luo[1,3,5], Dong-Sheng Yan[8], Chaoyu Zhang[2,5], Xiang Zhu [6], Qiuyue Yuan[2,5], Zhanying Feng [2,5], Chaoying Cui[7], Xuebin Qi [1,3], Ouzhuluobu[7], Wing Hung Wong [4,6,9 ✉], Yong Wang [2,3,5,10 ✉] & Bing Su [1,3 ✉]

High-altitude adaptation of Tibetans represents a remarkable case of natural selection during recent human evolution. Previous genome-wide scans found many non-coding variants under selection, suggesting a pressing need to understand the functional role of non-coding regulatory elements (REs). Here, we generate time courses of paired ATAC-seq and RNA-seq data on cultured HUVECs under hypoxic and normoxic conditions. We further develop a variant interpretation methodology (vPECA) to identify active selected REs (ASREs) and associated regulatory network. We discover three causal SNPs of EPAS1, the key adaptive gene for Tibetans. These SNPs decrease the accessibility of ASREs with weakened binding strength of relevant TFs, and cooperatively down-regulate EPAS1 expression. We further construct the downstream network of EPAS1, elucidating its roles in hypoxic response and angiogenesis. Collectively, we provide a systematic approach to interpret phenotype-associated noncoding variants in proper cell types and relevant dynamic conditions, to model their impact on gene regulation.

[1] State Key Laboratory of Genetic Resources and Evolution, Kunming Institute of Zoology, Chinese Academy of Sciences, 650223 Kunming, China. [2] CEMS, NCMIS, MDIS, Academy of Mathematics and Systems Science, Chinese Academy of Sciences, 100190 Beijing, China. [3] Center for Excellence in Animal Evolution and Genetics, Chinese Academy of Sciences, 650223 Kunming, China. [4] Bio-X Program, Stanford University, Stanford, CA 94305, USA. [5] University of Chinese Academy of Sciences, 100101 Beijing, China. [6] Departments of Statistics, Stanford University, Stanford, CA 94305, USA. [7] High Altitude Medical Research Center, School of Medicine, Tibetan University, 850000 Lhasa, China. [8] School of Mathematical Science, Inner Mongolia University, 010021 Huhhot, China. [9] Department of Biomedical Data Science, Stanford University School of Medicine, Stanford, CA 94305, USA. [10] Key Laboratory of Systems Biology, Hangzhou Institute for Advanced Study, University of Chinese Academy of Sciences, Chinese Academy of Sciences, 330106 Hangzhou, China. [11]Present address: Center for Human Genetics and Department of Genetics and Biochemistry, Clemson University, Greenwood, SC 29646, USA. [12]These authors contributed equally: Jingxue Xin, Hui Zhang, Yaoxi He, Zhana Duren, Caijuan Bai. ✉email: whwong@stanford.edu; ywang@amss.ac.cn; sub@mail.kiz.ac.cn

Tibetans have lived at high altitude (average 4000 m) for more than 30,000 years[1–3]. They survive the low oxygen environment by a distinct suite of physiologic traits: decreasing arterial oxygen content, increasing resting ventilation, lack of hypoxic pulmonary vasoconstriction, lower incidence of reduced birth weight, and relatively low hemoglobin (Hb) concentration. In contrast, Han Chinese lowlanders moving to high altitude develop increased Hb concentration to compensate for hypoxia but usually leads to polycythemia that increases the risk of heart attack, stroke, and fetal loss during pregnancy[4–7]. This blunted response to hypoxia in Tibetans is the result of natural selection acting on genes in oxygen intake, delivery, and utilization.

Previous genome-wide scans have identified many positively selected variants underlying high-altitude adaptation (Tibetan, Andean, and Ethiopian)[8–11]. In Tibetans, two major-effect genes (EPAS1/EGLN1) show the strongest signals of selective sweeps[7,12–14], which were significantly associated with a decreased hemoglobin phenotype. In addition, Tibetan-enriched EPAS1 variants were experimentally shown to down-regulate EPAS1's transcription[15], providing molecular basis at the transcriptomic level of Tibetans' blunted response to hypoxia.

However, major challenges remain to put those isolated data together and causally define genotype-phenotype relationships for high-altitude hypoxia adaptation. First, a great majority of these positively selected variants are located in non-coding regions which imply the critical role of gene regulation but challenges the elucidation of unknown cell type of action, relevant pathway, target gene, causality, and mechanism. In particular, the Tibetan-enriched EPAS1 variants are all non-coding ones, suggesting that gene expression regulation is likely selected for adaptation. In addition, these EPAS1 variants are highly linked and it is difficult to identify the causal variant(s) without context-specific multi-omics data. Second, hypoxia is a dynamic process and adaptation is a complicated phenotype involving many tissues. Functional variants are usually tissue-specific and may initiate the concerted effects of many regulatory elements and genes in a context-specific and time-dependent way. Third, accumulating evidence of existing studies suggests the polygenicity of hypoxia adaptation, i.e., high-altitude hypoxia adaptation is often affected by multiple genes/variants with small or moderate effects on transcription factors, regulatory elements and their target genes, and eventually trigger changes of a larger-scale regulatory network.

To address the above challenges, here we generate accessibility maps of REs across time series of hypoxia experiments in both Tibetan and Han Chinese and inferred hypoxia regulatory networks by linking these REs to their target genes (TG) and their transcription factor (TF) regulators. These maps and circuits allow us to understand how human genetic variants contribute to hypoxia adaptation.

## Results

**Chromatin accessibility landscape for hypoxia response and adaptation.** We design experiments to collect multi-omics data, in particular high-quality ATAC-seq data for chromatin accessibility landscape, to interpret the positively selected variants underlying high-altitude adaptation. Figure 1a shows our procedures for adaptive and wildtype population choosing, individual filtering by EPAS1/EGLN1 genotypes, HUVEC cell selection, time series hypoxia induction, multi-level omics data profiling, and quality control ("Methods" and Supplementary Fig. 1). We also confirm that response to being in culture for different periods of time are not significant compared to the responses to hypoxia ("Methods").

Multi-level omics data reveal multiple stages of hypoxia response. Hierarchical clustering of gene expression and chromatin accessibility indicates that hypoxia response is a multi-stage biological process (Fig. 1b). Gene expression profiles of 12,998 genes group 0 h/6 h/1 day as the first stage, and 3 day/5 day as the second stage. Chromatin accessibility profiles of 51,406 HUVEC enhancers show a three-stage landscape by further dividing the 0 h/6 h/1 day stage into 0 h sub-stage and 6 h/1d sub-stage. We identified 517 (out of 12,998 genes in total with proportion 3.98%) differential expressed genes (DEG) and 8551 (out of all 54,102 regions with proportion 15.81%) differential open regions (DORs) between normoxia 0 h and hypoxia 6 h with FDR < 0.05. The proportion of DORs is larger than DEGs by 4 folds. Overall, chromatin accessibility tends to respond earlier to hypoxia than gene expression. Unsupervised principal component analysis for genes and enhancers shows consistent patterns in larger variance under hypoxia response (PC1 and PC2) and small variance between populations. In addition, 6 h and 1 day cells tend to be similar under hypoxia pressure and 0 h, 3 day, and 5 day cells present large variation at both chromatin accessibility and gene expression levels (Fig. 1c). In contrast, the accessibility level shows a more smoothed and continuous trajectory than expression.

Genome-wide accessible regions allow us to identify transcription factors acting on regulatory elements to coordinately regulate gene expression and response to hypoxia. We identified 43 TFs with high expression of highly enriched motifs at different time points (Supplementary Fig. 3a). CTCF is highly expressed and enriched in all samples. This is consistent with its role to mediate proper looping between promoters and distal regulatory elements[16]. Other key regulators in endothelial gene expression, such as the E26 transformation-specific (ETS) family transcription factors (ERG, ETS1, FLI1, ELK4), together with AP-1, FOXO1, and GATA2[17] are identified. 14 TFs display interesting dynamics in motif enrichment and expression (Fig. 1d). JUN, ETS1, FOSL2, and TAL1 show a 6 h response, EPAS1 and MEF2D show a 1d response, and STAT6, KLF10, MAFK, and SOX17 show a 3d response to hypoxia. These early to late response patterns are consistent with their enriched GO processes, including response to oxidative stress, angiogenesis, and blood vessel/vasculature development (Supplementary Fig. 3b).

Genome-wide expression pattern shows that Tibetan samples have a blunted response to hypoxia as seen in the number of DEGs (Fig. 1e). DEGs are identified between adjacent time points by limma with FDR control of 0.05 ("Methods"). The pattern again shows that hypoxia response is a multi-stage process with more than 8,000 DEGs between 1d to 3d. Decline of DEGs over a five-day period is consistent with our previous report[15]. This decline pattern is likely caused by the known regulatory feedback loops in hypoxic response[18]. Moreover, Tibetans have fewer DEGs than Han in all four adjacent time point comparisons (348, 1068, 537, and 565 fewer DEGs with proportions of total DEGs 40.23%, 83.31%, 6.62%, and 18.24%). This indicates Tibetans' response to hypoxia with less dramatic gene expression changes than Han, i.e., the blunted effect.

Importantly, 111,182 positively selected variants for high altitude adaptation are enriched in the open regions of our 50 HUVEC samples with varying thresholds ("Methods" and Supplementary Fig. 4b). Open chromatin regions under hypoxia and open regions under normoxia both show over 1.5 fold enrichment, which are significantly different from open regions in ESC (embryonic stem cell) as control (DNase-seq sample in ROADMAP) (Fig. 1f). The SNPs with signals of positive selection tend to have 20% more enrichment in hypoxia (6 h) than in normoxia (0h). This pattern is consistent for the 50 samples (Supplementary Fig. 4b). HUVEC shows relatively high

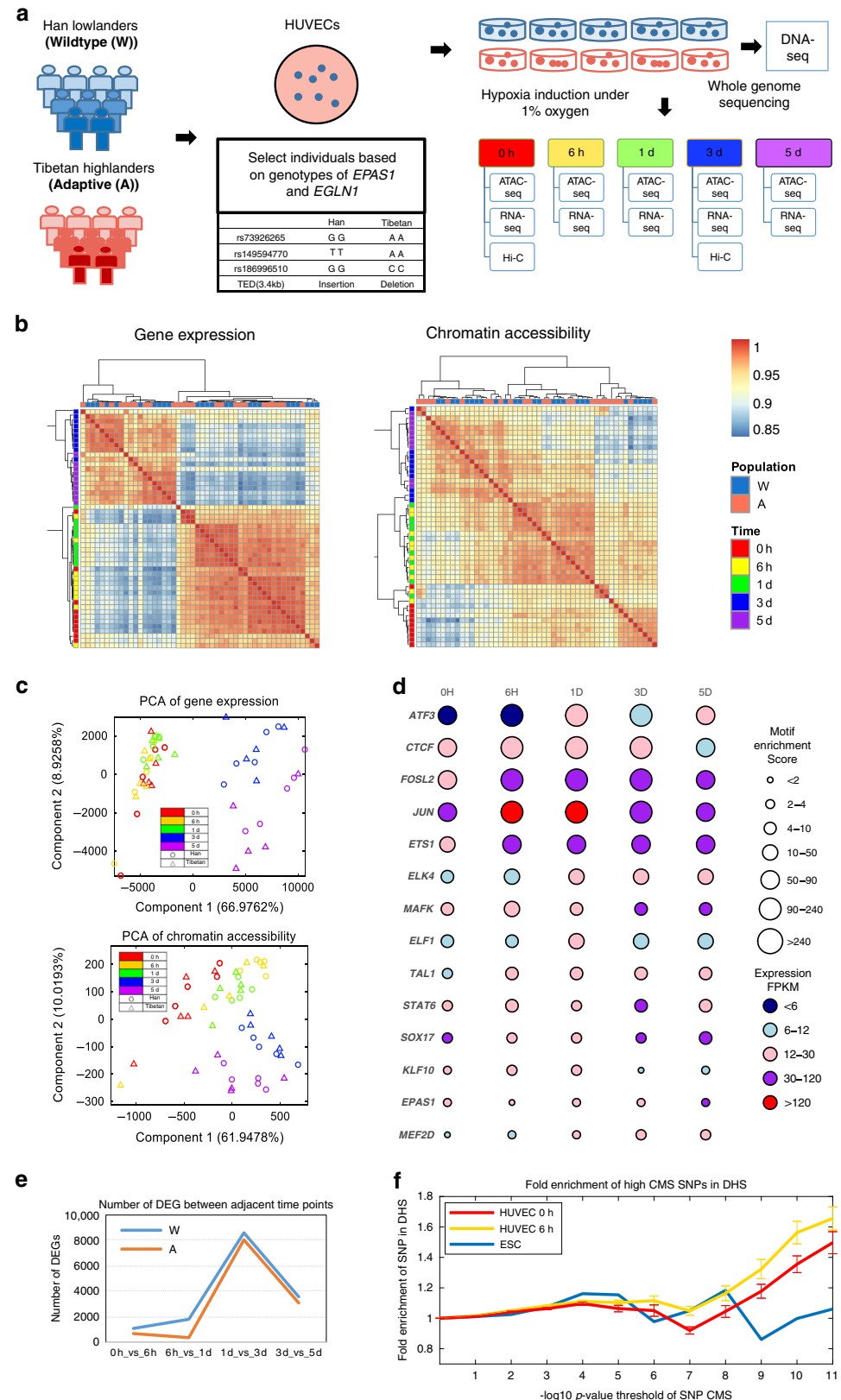

enrichment compared with a large panel of cell types from ROADMAP (Supplementary Fig. 4c).

**Genome-wide statistical modeling and data integration.** We propose a statistical model, vPECA (Variants interpretation

model by Paired Expression and Chromatin Accessibility data), to Integrate our measured paired expression and chromatin accessibility data with the available public data, including population genetics data, functional genomics data in ENCODE, and Hi–C data for HUVEC. Our previous work PECA integrates paired expression and chromatin accessibility data across diverse cellular

**Fig. 1 Paired expression and chromatin accessibility time-series data reveal the regulatory landscape for high altitude adaption. a** Experimental design diagram for adaptive and wildtype population choosing, individual selection by *EPAS1/EGLN1* genotypes, HUVEC cell culture, time-series hypoxia induction, and multi-level omics data profiling. **b** Hierarchical clustering of gene expression (left panel) and chromatin accessibility (right panel) indicates hypoxia is a multi-stage biological process. Gene expression profiles group 0 h, 6 h, and 1 day as the first stage, and 3 day and 5 day as the second stage. Chromatin accessibility responses earlier to hypoxia than gene expression by further dividing the first stage into two sub-groups (0 h and 6 h, and 1 day). **c** Unsupervised principal component analysis for 12,998 genes (upper panel) and 51,406 HUVEC enhancers (lower panel). In all, 6 h and 1 day cells tend to be similar under hypoxia pressure and 0 h, 3 day, and 5 day samples present large variation at both chromatin accessibility and gene expression. Accessibility pattern shows a more smoothed trajectory than expression. **d** TF response to hypoxia by their enriched motifs and gene expression dynamics across time points. **e** Tibetan samples show the blunted responses to hypoxia by the number of differentially expressed genes (DEGs) between adjacent time points. DEGs are identified by limma with FDR control 0.05. **f** Positively selected SNPs for high altitude adaptation quantified by the CMS score thresholding by $-\log_{10}(p\text{-value})$ are enriched in the open regions revealed by HUVEC ATAC-seq data but not in embryonic stem cells (ESC). *P*-value of CMS score for each SNP were calculated by Fisher's method ("Methods" for details). Error bars indicate the mean ± standard error of fold change between replicates ($n = 10$ biologically independent samples). Source data are provided as a Source data file.

contexts and model the localization to REs of chromatin regulators (CR), the activation of REs due to CRs that are localized to them, and the effect of TFs bound to activated REs on the transcription of target genes (TG)[19]. Our innovation here is to extend PECA to interpret genetic variants from population genetics and matched WGS data. vPECA models how positively selected noncoding SNPs affect the RE's selection status, chromatin accessibility, and activity and further determine the target gene expression in relevant cellular context (Fig. 2a). The statistical modeling allows us to systematically identify active REs, active selected REs, and gene regulatory network to interpret variants.

We prepare the genome annotation of all transcriptional units (genes), regulatory elements (REs) and high-resolution 3D chromatin interactions in HUVEC, and positively selected variants with various quantitative scores and their LD associations ("Methods" and Fig. 2a). Table 1 summarizes the types of data to be analyzed or incorporated into our model of gene regulation. To model gene regulation with RE activity and selection, our analytical approach learns from this data to generate the distribution of the expression of target genes conditional on the accessibility of regulatory elements and the expression of transcription factors. Our model, depicted in Fig. 2a, has four components designed to model respectively (1) control of a target gene expression; (2) activity status of a regulatory element; (3) selection status of a regulatory element, and (4) the effect of RE's selection status on accessibility.

*Expression of target gene*: we assume that the rate of transcription of a TG in a cellular context is affected by TFs bound to regulatory elements that are active in that cellular context. For each RE we construct a variable (parenthesized term in Eq. (4) of Fig. 2a) that represents the combined effect of TFs that are expressed in that context and have significant motif matches on that RE. Target gene expression is modeled by regression with these variables as potential predictors. However, only active REs associated with a TG will be included in the regression model for that TG (Fig. 2a and Eq. (3)). The association of RE to TG was restricted by the Hi–C TAD loop boundary and the degree of correlation between the accessibility of the RE with promoter accessibility and expression of the TG across tissues by PECA (Methods).

*Activity status of regulatory element*: the activity status of a RE (say the *i*th RE) is represented by a context-dependent variable $Z_i$, indicating the active state of *i*th RE. The knowledge of the selection status of a RE is informative on the activity status of that RE. To incorporate this into our model, we denote the activity status of a RE by a binary variable *S*, i.e., $S_k = 1$ indicates that the *k*th RE is under positive selection. These variables are used together with the accessibility of the RE, as predictive variables in our model for the activity status of the RE (Fig. 2a and Eq. (3)).

*Selection status of RE*: we assume a RE is likely to be selected if the RE contains positively selected SNPs. We integrated multiple SNPs in a RE by down-weighting the effect of LD structure. In addition, multiple selection scores are combined to assess the positive selection of SNPs in order to balance their advantages and disadvantages. The resulting model for selection status prediction is given in Eq. (1) of Fig. 2a.

*Effect of RE's positive selection on accessibility*: regulatory mutations can drive chromatin accessibility changes in direct or indirect ways[20]. For example, a non-coding variant causes the generation of TF binding site, this variant could lead to an increase of chromatin accessibility in a cis-RE and concomitant increase in the observed frequency of the mutant allele. Or indirectly, the variant will impact the 3D chromatin interactions and increase accessibility. If this variant in RE is functional and can be fixed by natural selection, we expect the chromatin accessibility increase or decrease will be associated with the RE's selection status. The resulting model for accessibility effect is given in Eq. (2) of Fig. 2a.

We propose an algorithm to infer the unknown parameters $\mu$, $\alpha$, $\beta$, $\gamma$, $\omega$, $\sigma^2$ and latent variables ($S$, $Z$) based on the observed expression data ($TG$, $TF$), accessibility data ($O$), and the selection and LD status of SNPs ($X$, $Y$). We consider the conditional density of $TG$ and $O$ given $TF$, $X$, $Y$, and $O$:

$$P(TG, O|TF, O, X, Y) = P(O|TF, X, Y)P(TG|TF, O, X, Y)$$
$$= \sum_{Z,S} P(TG|TF, Z)P(Z|S, O)P(O|S)P(S|X, Y)$$

The term $P(S|X, Y)$ represents the conditional density of the selection status of the *t*th RE, as specified Eq. (1) of Fig. 2a. Similarly the terms $P(O|S)$, $P(Z|S, O)$, and $P(TG|TF, Z)$ are specified by Eqs. (2)–(4) of Fig. 2a. Note that these terms involve different components of the parameter vector: $\mu$ appears in the first term, $\omega$ appears in the second term, $\alpha_t$ appears in the third term, and ($\beta_i$, $\gamma_k$) appears in the fourth term. This conditional experiment ($TG$, $O|TF$, $O$, $X$, $Y$) provides a valid basis for the inference of the unknown parameters $\mu$, $\alpha$, $\beta$, $\gamma$, $\omega$, $\sigma$ and latent variables ($S$, $Z$). To induce sparsity, we use Laplacian priors for the parameters $\alpha$, $\beta$, and $\gamma$. We employ an iterated conditional mode algorithm for this inference. The resulting model of inference methodology is named vPECA, extending our previous Paired Expression and Chromatin Accessibility modeling[19] (Table 1 and "Methods").

**Active and selected REs and their regulatory targets.** vPECA identifies 32,330 active REs (ARE) ($\beta \neq 0$) for 9952 genes including 1647 active selected REs (ASRE) ($P(S = 1) > 0.95$, $\alpha \neq 0$ and $\beta \neq 0$) for 1146 genes (Supplementary Data 2) and the associated regulatory network including 52,647 interactions among REs and target genes. Each gene is on average regulated by

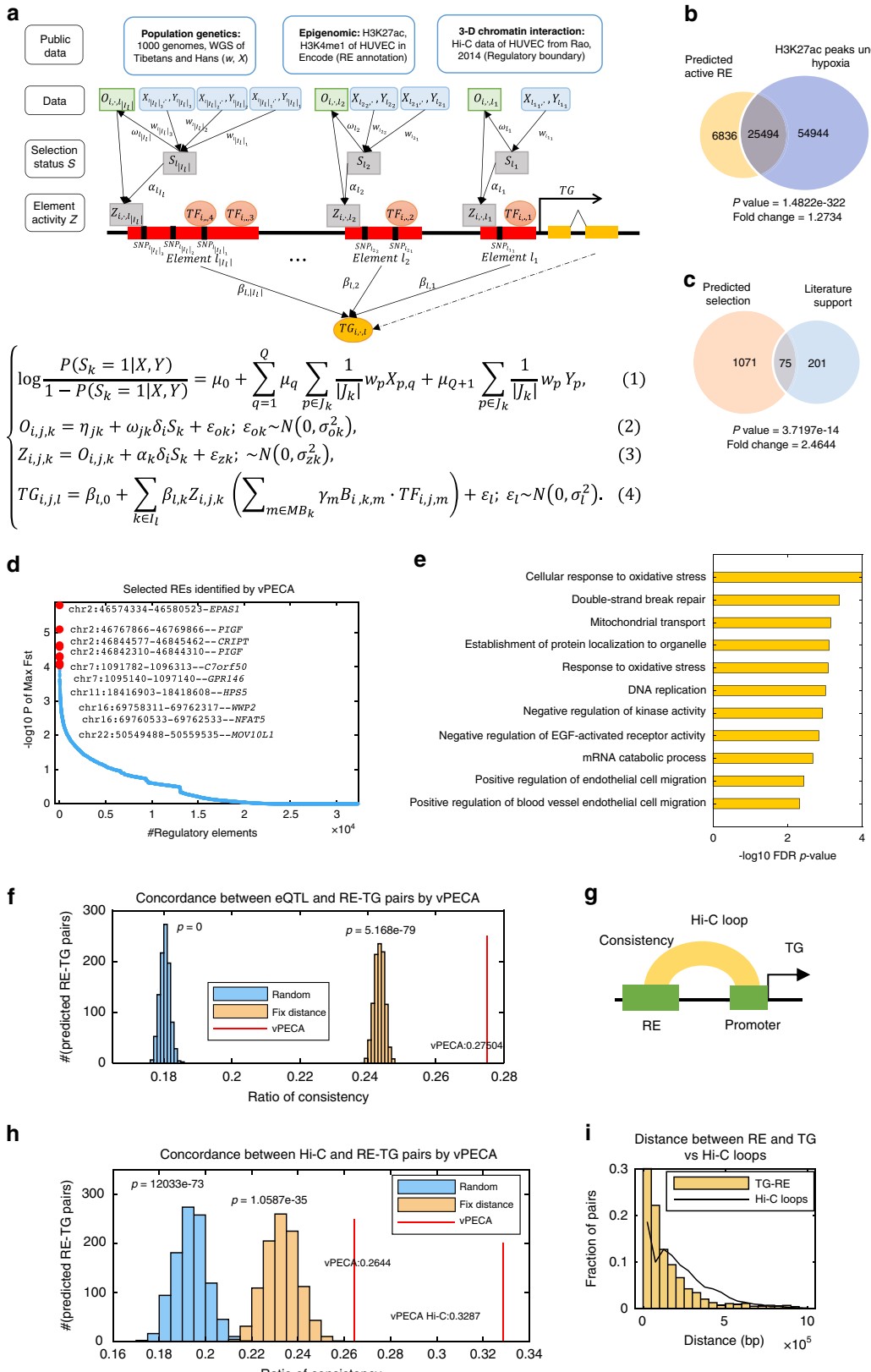

$$\log \frac{P(S_k = 1 | X, Y)}{1 - P(S_k = 1 | X, Y)} = \mu_0 + \sum_{q=1}^{Q} \mu_q \sum_{p \in J_k} \frac{1}{|J_k|} w_p X_{p,q} + \mu_{Q+1} \sum_{p \in J_k} \frac{1}{|J_k|} w_p Y_p, \quad (1)$$

$$O_{i,j,k} = \eta_{jk} + \omega_{jk} \delta_i S_k + \varepsilon_{ok}; \ \varepsilon_{ok} \sim N(0, \sigma_{ok}^2), \quad (2)$$

$$Z_{i,j,k} = O_{i,j,k} + \alpha_k \delta_i S_k + \varepsilon_{zk}; \ \sim N(0, \sigma_{zk}^2), \quad (3)$$

$$TG_{i,j,l} = \beta_{l,0} + \sum_{k \in I_l} \beta_{l,k} Z_{i,j,k} \left( \sum_{m \in MB_k} \gamma_m B_{i,k,m} \cdot TF_{i,j,m} \right) + \varepsilon_l; \ \varepsilon_l \sim N(0, \sigma_l^2). \quad (4)$$

five active REs (Supplementary Fig. 5a). In addition, we use $\omega \neq 0$ to select those REs under selection with differential accessibility in local time points. Target genes are enriched for functions in stress response and metabolic process etc. (Supplementary Fig. 5b).

We validate the selection status, active status, and inferred RE-TG interactions using independent data resources. The active REs revealed by vPECA are significantly enriched in H3K27ac peaks exposed to the 1% hypoxia environment for 24 h from an independent public data source[21]. In all, 25,494 predicted REs are

**Fig. 2 vPECA systematically reveals active REs, positively selected active REs, and regulatory network. a** vPECA models how positively selected noncoding SNPs affect the RE's selection status, chromatin accessibility and activity, and further determine the target gene's (TG) expression by integrating paired expression and chromatin accessibility profiling with population genetics, epigenome, and 3D chromatin interaction data. Refer to Table 1 for model components and notations and "Methods" for details of vPECA). **b** Active REs revealed by vPECA are significantly enriched in H3K27ac peaks exposed to 1% hypoxia environment for 24 h from an independent public data source (Fisher's exact test). **c** Active selected REs associated genes are enriched in the literature reported high altitude adaption gene. *P*-value is calculated by hypergeometric test. **d** Active selected REs are ranked by their selection (maximum Fst score of all SNPs in the given RE) and chromatin accessibility change score. **e** The active selected REs are enriched in biological processes related to hypoxia adaptation. *P*-values were calculated by hypergeometric test with Benjamini–Hochberg correction. **f** eQTL data in the GTEx database significantly overlapped with the vPECA identified RE-TG pairs. The null distribution by a random selection of the same number of REs nearby expressed genes for 1000 times. "Fix distance" means only selecting TG-RE pairs with the same distance distribution as eQTL data. *P*-values were calculated by one-sided *t*-test. **g** A RE-TG pair is validated if their RE and promoter are linked by at least one Hi-C loop in hypoxia or normoxia. **h** HiC loop under hypoxia significantly overlapped with the vPECA identified RE-TG pairs. The null distribution is constructed by the random selection of the same number of REs nearby expressed genes in HUVEC for 1000 times. "Fix distance" means only selecting TG-RE pairs with the same distance distribution as Hi–C loop. Sensitivity is calculated by the number of RE-TG pairs validated by Hi–C loops normalized by the total number. *P*-values were calculated by one-sided *t*-test. **i** vPECA tends to link proximal REs to the promoter, while Hi–C loops detect distal interactions. Source data are provided as a Source data file.

**Table 1 vPECA model component and notations.**

| Data and variables | Notations | Examples |
|---|---|---|
| *Individual and time-dependent data* | | |
| Expression of TF | $TF_{i,j,m}$: = expression of the *m*th TF of individual *i* on time *j* | $TF_{W1,1,EPAS1} = 23.87$ |
| Expression of TG, not TF | $TG_{i,j,l}$: = expression of the *l*th TG of individual *i* on time *j* | $TG_{W1,1,NQO1} = 97.35$ |
| Accessibility of RE | $O_{i,j,k}$: = openness of the *k*th RE of individual *i* on time *j* | $O_{W1,1,chr2:46589710-46594828,} = 1.14$ |
| *Individual and time dependent latent variable* | | |
| Activation status of a RE | $Z_{i,j,k}$: = activation status of the *k*th RE of the *i*th individual on time *j* | $Z_{W1,1,chr2:46574334-46580523} = 2.72$ |
| *Population dependent data* | | |
| $F_{ST}$ score | $X_{p,1}$: = $F_{ST}$ score of *p*th SNP | $X_{rs3768729,1} = 4.62$ |
| iHS score | $X_{p,2}$: = iHS score of *p*th SNP | $X_{rs3768729,2} = 9$ |
| XP-EHH score | $X_{p,3}$: = XP-EHH score of *p*th SNP | $X_{rs3768729,3} = 1.94$ |
| PBS score | $X_{p,4}$: = PBS score of *p*th SNP | $X_{rs3768729,4} = 4.94$ |
| Weight of the *p*th SNP | $w_p$: = weight of the *p*th SNP derived from LD score | $w_{rs3768729} = 0.0583$ |
| TFs with motif on a RE | $MB_k$: = set of TFs with motif match in *k*th RE | ARNT has motif match at chr2:46589710-46594828 |
| *Individual dependent data* | | |
| The difference of Derived Allele Frequencies between two populations ΔDAF | $Y_p$: = ΔDAF of *p*th SNP | $Y_{rs3768729} = -0.8$ |
| Motif matching strength of TF on RE | $B_{i,k,m}$: = matching strength of the *m*th TF binding on the *k*th RE of individual *i* | $B_{W1,chr2:46589710-46594828,BACH1} = 2.90$ |
| *Population and individual dependent latent variables* | | |
| Selection status of a RE | $S_k$: = selection status of the *k*th RE | $S_{chr2:46589710-46594828} = 1$ |

validated (78%) with Fisher's exact test *p*-value $1.48 \times 10^{-322}$ and fold change 1.27 (Fig. 2b).

vPECA identified 1146 genes associated with ASREs and they are significantly overlapped with 549 literature reported high altitude adaptation genes for Tibetans (only 276 are expressed in HUVEC)[10] with hypergeometric test *p*-value $3.71 \times 10^{-14}$ and fold change 2.46 (Fig. 2c). We note that selected genes are reported when there are high Fst SNPs in coding regions or inside the gene body region. We further checked the 1,071 selected genes identified by vPECA (Fig. 2c, genes without literature support) and found that the vPECA model could integrate many weakly selected SNPs and downstream accessibility and expression change as well as assign strongly selected SNPs to a distal gene. Anti-oxidant enzyme *NQO1* is reported as a selected gene by vPECA due to 48 weakly selected SNPs with Fst score about 0.15 (Supplementary Fig. 5c and large accessibility and expression change (Supplementary Fig. 5d). Cell mobility associated gene *ACTG1* is regulated by two SNPs (Fst > 0.23) 55Kb upstream and

shows the advantage of assigning distal target genes to high Fst SNPs by the long-range RE-gene interactions inferred by vPECA (>10 kbp) (Supplementary Fig. 5e).

Active selected REs are ranked by their selection (the maximum Fst score of all SNPs in a given RE) and chromatin accessibility change score. RE "chr2_46574334_46580523" in the intron region of *EPAS1* has the highest rank (Fig. 2d). This demonstrates that vPECA can infer selection status for non-coding REs. Functional enrichment analysis for the active selected REs reveals the key process related to hypoxia adaptation, such as "response to oxidative stress", "mitochondrial transport", and "positive regulation of blood vessel endothelial cell" (Fig. 2e).

In addition, vPECA predicts RE-gene interactions via cross-sample activity correlation. This functional evidence should be complementary with the genetic evidence via SNP effect on gene expression and physical evidence via proximity in 3D. eQTL can provide enhancer-gene interaction in a tissue-specific way. We used the eQTL data from GTEx to overlap with our

predicted active RE-gene interactions. Totally 65,732 eQTL interactions across 44 cellular contexts are collected ("Methods"). Overall 18,079 out of 52,647 predictions have eQTL support such that they overlap at least one reported eQTLs. This is significantly larger than expected with one-sided $t$-test $p$-value $< 10^{-70}$ (Fig. 2f). The overlaps in 44 individual tissues are all significant (Supplementary Fig. 5f). Those tissues sensing oxygen similar to HUVEC tend to have a higher ratio of consistency. We further checked the concordance between chromatin loops from the Hi–C experiment and RE-TG pairs identified by vPECA (Fig. 2g). 33% of Hi–C loops are concordant with at least one RE-TG pair identified by vPECA. To assess the significance, we generated the null distribution by a random selection of the same number of REs in HUVEC nearby expressed genes for 1000 times. vPECA gives 2-fold higher than randomly selected pairs with one-sided $t$-test $p$-value $1.2 \times 10^{-73}$ (Fig. 2h). If the null distribution is generated with physical distance constraint, i.e., the distance between randomly selected RE-TG pairs follows the same distribution with vPECA identified ones, vPECA still shows significantly higher enrichment with one-sided $t$-test $p$-value $1.06 \times 10^{-35}$. To exclude the potential bias due to public normoxia HUVEC Hi–C, we re-run the vPECA model without using Hi–C data to define physical boundaries and still 26% Hi–C loops are concordant with vPECA prediction (one-sided $t$-test, $p$-value $1.2669 \times 10^{-24}$). The distribution of RE-TG distance shows that vPECA tends to link proximal REs to promoter and complement to Hi–C loops which only detect distal interactions (Fig. 2i).

**The mechanism for down-regulation of *EPAS1* in Tibetans' hypoxia response.** *EPAS1* is the most important hypoxia-inducible transcription factor and shows the strongest selective sweep in Tibetans. Our evolutionary omics strategy helps explore the *EPAS1* regulatory mechanism in hypoxia and adaptation. vPECA successfully identifies *EPAS1*'s upstream active REs and explains its down-regulation mechanism in expression. *EPAS1*'s regulatory map includes annotated REs, positively selected SNPs, Hi–C loops restricting the regulatory boundary (Fig. 3a). In all, 23 potential REs are annotated by epigenomic data (DNases-seq, H3K27ac, H3K4me1, and H3K4me3). Positively selected SNPs are picked by their Fst scores from population genetics study[8]. LD structures are derived from Tibetan populations[8] and 1000 Genomes and the positively selected SNPs (Fst > 0.5) show more compact linkage in Tibetans.

By integrating the static regulatory map with time-series hypoxia omics data, vPECA identified 7 active REs potentially regulating *EPAS1*'s expression in HUVEC (Fig. 3a and Supplementary Data 3). Three elements within introns of *PRKCE* (E7, E8, and E12) are predicted to regulate *EPAS1* because they locate in the same Hi–C loops with the promoter of *EPAS1* but not *PRKCE* (Supplementary Fig. 6a). From the REs' dynamics pattern in chromatin accessibility (Supplementary Fig. 6b), 5 REs (E7, E8, E12, E17, and E21) regulate *EPAS1* across all time points and populations ($\beta \neq 0$). 2 REs (E20, E22) show significant interactions between population and expression dynamics ($\omega \neq 0$). E20 responses to hypoxia earlier and E22 responses quite late. From the inferred selection status, we group the REs into two types: (A) 3 REs (E7, E8, E12) as active REs and (B) 4 REs (E17, E20, E21, E22) as active selected REs (Supplementary Data 3).

Under the regulation of those REs, we observed *EPAS1*'s expression level increased by 30% after 1 day hypoxia treatment. It is significantly less expressed (30% reduction) in Tibetan than in Han (two-sided $t$-test $p$-value is 0.002) (Fig. 3b). This confirms the *EPAS1*'s down-regulation pattern at expression level[15], which

contributes to the molecular basis of Tibetan's blunted response to high-altitude hypoxia.

Chromatin accessibility dynamics well explain *EPAS1*'s expression. Linear regression of *EPAS1*'s expression by 23 REs' accessibility (Supplementary Data 3) identifies 4 REs (E2, E12, E20, E21) having nonzero coefficients. Two active REs (E12, E21) can explain 54% variance. E12 contributes the most by 34%, and the percentage increases to 54% when adding E21 (Supplementary Fig. 6c). This is due to the fact that both E12 and E21 show a global accessibility pattern correlated with expression (Fig. 3c). Utilizing the binding TFs on the REs predicted by vPECA, TFs and REs together can increase the percentage to 68% (Supplementary Fig. 6d).

We used Hi–C data to test if RE-promoter physically interacts. vPECA predicted active REs of *EPAS1*, i.e., E7, E8, E12, E17, E20, E21, and E22, show physical interactions with the *EPAS1*'s promoter in hypoxia d3 and the interactions are enhanced from normoxia to hypoxia (Fig. 3d). We reversely queried RE's interacting regions and the promoter shows the strongest signal (Supplementary Fig. 7). We further applied dual-luciferase enhancer reporter gene assay in both normoxic and hypoxic conditions to test if the five active REs (E2, E12, E20, E21, and E22) functionally affect the expression of the downstream gene. Assays transfected to two independent cell lines HEK293 and HELA (each with three replicates) were cultured in normoxia and hypoxia (1% oxygen) environments for 36 h (Fig. 3e), and an empty vector was used as control. All the five active REs (E20, E21, E22 with adapted and wildtype alleles, E2, and E12) show significantly higher activities in hypoxia than in normoxia (two-sided $t$-test $p$-values < 0.05) in at least one cell line, except for E21-2. Together, the identified *EPAS1*'s REs physically interact with its promoter in the Hi–C data and are likely functional.

**Interpretation of causal SNPs that regulate *EPAS1*.** vPECA reveals two types of REs to achieve a fine-tuned down-regulation of *EPAS1* by modeling epigenomic data to interpret high Fst SNPs. One type is active REs without selection, such as E2 and E12. The second type is the active REs with strong selection, such as E20, E21, and E22. The three selected active REs, E20, E21, and E22, tend to show significant chromatin accessibility difference between Tibetan (adaptive) and Han (wildtype) globally or locally across time points (Fig. 3c). The two types of REs work together to regulate *EPAS1*'s expression and the combined effect of E12 and E21 is revealed in a stepwise linear regression (E12 explains 30% and E21 explains an additional 20%) (Supplementary Fig. 6c). Furthermore, these two REs tend to be clustered in the H3K27ac peak regions across tissues and their activities positively correlate with EPAS1's expression (Supplementary Fig. 6e).

vPECA used the predicted active REs to interpret SNPs' causal effect on gene expression. We experimentally tested five SNPs with high Fst scores located in E20, E21, and E22. The presumably adaptive alleles are "C" for rs569774785; "G, A" for rs4953357, rs6756667; "A, G" for rs10206434, rs141366568; "A, T" for rs370299814, rs368706892 and "T" for rs3768729, and all SNPs associate with RE's chromatin accessibility changes (Supplementary Fig. 8a). For SNP rs3768729, the putatively adaptive allele is ancestral and it has a large allelic divergence between Tibetan and other populations, especially for East Asian populations (Fst = 0.522 for CHB; Fst = 0.574 for JPT) (Supplementary Fig. 8b). Strikingly, the dual-luciferase enhancer reporter gene assay around rs3768729 located in E22 (E22-1) clearly indicates a weaker activity of the Tibetan-enriched "T" allele compared with the wild-type "C" allele in both cell lines and under both conditions (Fig. 3h). The two-sided t-test p-values are $2.18 \times 10^{-6}$ (HEK293 in normoxia), $2.40 \times 10^{-6}$ (HELA in

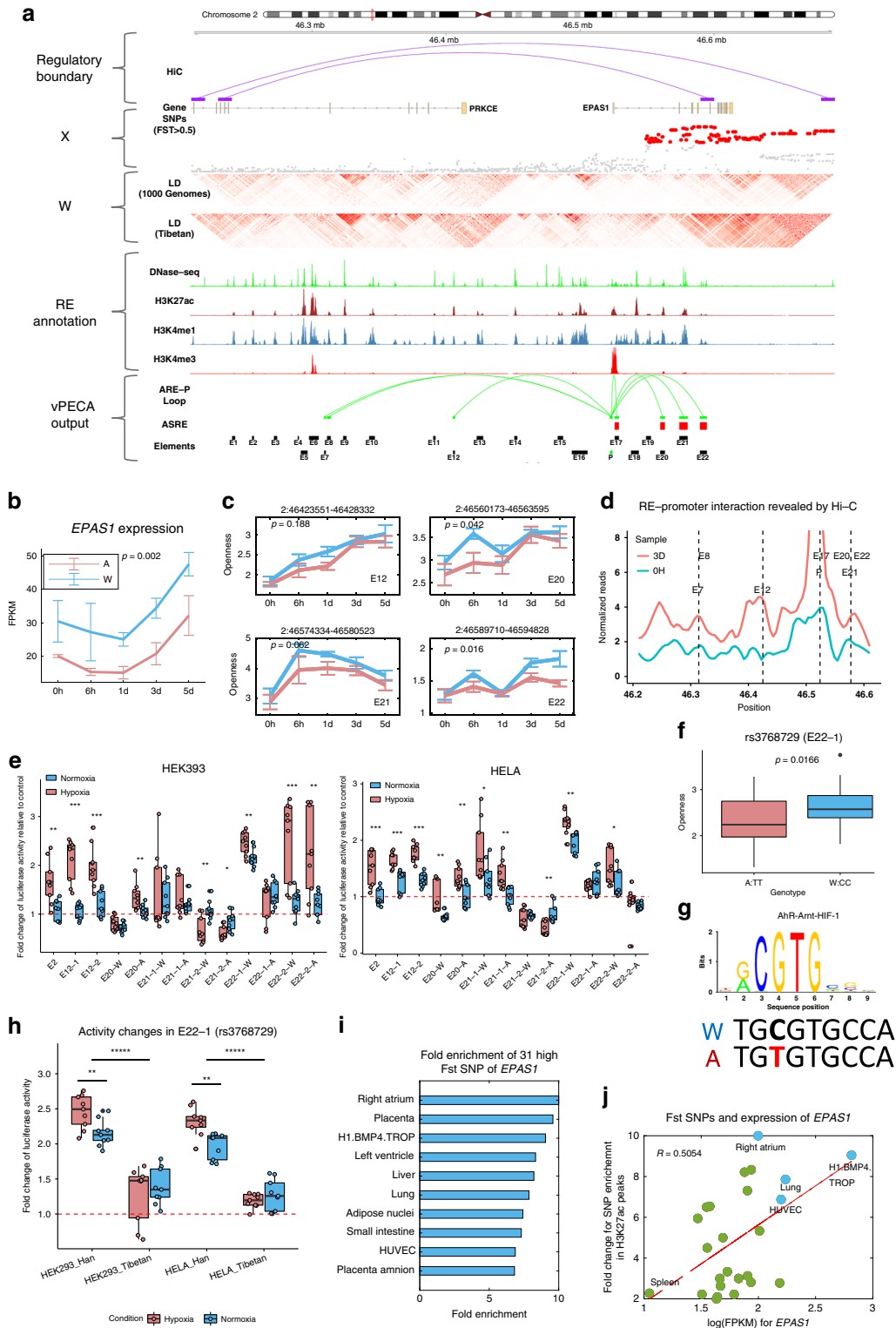

normoxia), $1.18 \times 10^{-6}$ (HEK293 in hypoxia), and $9.35 \times 10^{-11}$ (HELA in hypoxia), respectively. Notably, under hypoxia, the activity differences for E22-1 between Han and Tibetan became larger (two-sided t-test, $p = 0.0038$ in HEK293 and $p = 3.51 \times 10^{-5}$ in HELA). The "T" allele also shows significantly higher accessibility in the $[-200\ \text{bp}, +200\ \text{bp}]$ region around rs3768729 in Han than in Tibetan (one-sided t-test, p-value 0.0166) (Fig. 3f). This SNP may weaken the binding strength of the AhR-Arnt-HIF-1 complex (Transfac motif ID: MA0259) to E22 in Tibetan

(Fig. 3g). It changes the first position of mandatory core consensus CGTG in the high-quality motif derived from oxygen-regulated elements of 70 known HIF target genes[22–24], which partly explain the strong effect in the reporter assay. Together, we have the evidence that single SNP will change motif binding and chromatin accessibility of E22, alter E22's activity, and eventually decrease *EPAS1*'s expression. This causality has been strongly supported by the reporter assays in two independent cell lines (HEK293 and HELA), implying that the

**Fig. 3 EPAS1's upstream active REs explains its down-regulation mechanism. a** Regulatory map for *EPAS1* shows Hi–C loops restricting the regulatory boundary, epigenomic data (DNases-seq, H3K27ac, H3K4me1, and H3K4me3) annotating the 23 REs, and SNPs under positive selection with their Fst score and LD structure (from Tibetan populations and 1000 Genomes). vPECA outputs active RE and regulations. **b** *EPAS1*'s down-regulation pattern at the expression level. *P*-value was calculated by two-sided *t*-test ($n = 25$). Data are presented as mean values ± standard error ($n = 3$). **c** Chromatin accessibility dynamics of *EPAS1*'s four active REs, E12, E20, E21, and E22. *P*-values were calculated by two-sided *t*-test ($n = 25$). Data are presented as mean values ± standard error ($n = 5$). **d** *EPAS1*'s RE-*EPAS1* regulations are validated by HiC's chromatin interaction data between REs and promoter. **e** Dual-luciferase enhancer reporter gene assay of five upstream *EPAS1* variants in normoxic and hypoxic culture conditions of 37 °C in two cell lines HEK293 (left) and HELA (right). All assays were performed in three independent experiments (each with three technical replicates, $n = 9$). The p-values were calculated by two-sided *t*-test (*$P < 0.05$; **$P < 0.01$; ***$P < 0.001$). A indicates adaptive and W for wildtype. **f** The $[-200 \text{ bp}, +200 \text{ bp}]$ region around rs3768729 shows differential accessibility between Tibetan and Han samples. The p-values were calculated by one-sided *t*-test ($n = 25$ samples). **g** The SNP rs3768729 weakens TF complex HIF1A-ARNT (AhR,-Arnt,-HIF-1_transfac_M00976)'s binding strength in Tibetan. **h** Dual-luciferase enhancer reporter gene assay of E22-1 (rs3768729). All assays were performed in three independent experiments (each with three technical replicates, $n = 9$) and the *p*-values were calculated by two-sided *t*-test (**$P < 0.01$; *****$P < 0.00001$). **i** *EPAS1*'s 31 high Fst SNPs are enriched in the enhancers in other tissues defined by H3K27ac. **j** The number of *EPAS1*'s high Fst SNPs in REs is positively correlated with *EPAS1*'s expression level across tissues. The fold change is defined as the number of high Fst SNPs per kb in context active region. All boxplots in this figure are represented by minima, 25% quantile, median, 75% quantile, and maxima. Source data are provided as a Source data file.

Tibetan-specific SNP is likely functional (Fig. 3h). Similarly, the $[-200 \text{ bp}, +200 \text{ bp}]$ region around rs141366568 in E22 shows differential accessibility between Tibetan and Han (*p*-value $3.62 \times 10^{-6}$) (Supplementary Fig. 8a). The SNP rs141366568 (in E22-2) may causally weaken SOX17's binding strength in E22 (Supplementary Fig. 8c), repress *EPAS1*'s expression, which was validated by the reporter assay in HELA and HEK239, and the element activity in hypoxia is significantly larger than in normoxia (Fig. 3e). The $[-200 \text{ bp}, +200 \text{ bp}]$ region around rs569774785 in E21 shows differential accessibility between Tibetan and Han (one-sided *t*-test, *p*-value $< 10^{-4}$, Supplementary Fig. 8a). The SNP rs569774785 (in E21-1) may causally weaken the receptor RORA's binding strength in E21, repress EPAS1's expression and this was validated by the reporter assay in HELA with higher activity in hypoxia (Fig. 3e and Supplementary Fig. 8d). We noted that a SNP in *RORA* is the second strongest signal of association with Hb concentration in the Amhara in Ethiopia and directly regulates *HIF1A*[25,26]. Given *EPAS1*'s strongest selection signal and association to hemoglobin concentration in Tibetan adaptation, this *EPAS1*'s potential regulator RORA may imply the convergent evolution of hemoglobin regulation in high altitude adaptation.

In Supplementary Fig. 6b, *EPAS1* is regulated by multiple enhancers, whose activity is different along time and affected by various SNPs (Fig. 3e) in the tightly linked haplotype. Although the functions and effect sizes of these SNPs and enhancers might be different, some even have an inverse pattern, the overall output is a down-regulation of *EPAS1* in Tibetan compared with Han Chinese.

In addition to the 3 causal SNPs, we observed that many positive selected SNPs are not in HUVEC's active REs (Fig. 3a). A possible explanation is that those SNPs may be functional in other tissues (Fig. 3i). We overlap the set of 31 SNPs with the largest allelic divergence (Fst > 0.5) between Tibetan and Han[15] with predicted enhancers (H3K27ac peaks) in the 127 ROAD-MAP cell types. The enrichment is assessed for the overlap with enhancers in each cell type by comparing with two background models: all 1000 Genomes variants with a frequency above 5% in any population and all independent GWAS catalog SNPs. The enrichment relative to these background frequencies is performed using a binomial test and fold change. The analyzed key tissues include the right atrium, placenta, and H1 BMP4 derived trophoblast cultured cells, etc.

We demonstrate that the regulatory mechanism of three positively selected SNPs is to reduce *EPAS1*'s expression level. This allows us to hypothesize that the higher *EPAS1*'s expression level, the more difficult is the task and more selected SNPs are

required. Indeed, we found that the number of high Fst SNPs in context-specific H3K27ac region associated with *EPAS1* is positively correlated with *EPAS1*'s expression level across tissues (with Pearson correlation coefficient (PCC) 0.505) (Fig. 3j). The higher the expression in the tissue, the higher the fold enrichment of the large Fst SNPs in their H3K27ac peaks. For example, HUVEC utilizes many high Fst SNPs in its active REs to regulate *EPAS1*'s high expression in contrast to ESC (Supplementary Fig. 8e). The plot implies the key tissues for *EPAS1*'s function are the right atrium, lung, HUVEC, and H1 BMP4 derived trophoblast cultured cells. Given the fact that *EPAS1* shows the strongest selection, we predict these tissues contain the most relevant cell types for high altitude adaptation. We further hypothesize that this is a general mechanism and plot the nine genes with PCC larger than 0.5 in Supplementary Fig. 8f, including *LDHA* and *NEK7*, which are identified under selection in Tibetan and highly expressed in hypoxia-related cell types ("Methods").

**Reconstruction of regulatory network downstream of *EPAS1*.** *EPAS1* is the major-effect gene in Tibetan's high-altitude adaptation and is the master TF regulator for development and many processes. We extract all the active REs with EPAS1's motif binding and link those REs with TGs, and pool all the EPAS1-RE-TG triplets (Fig. 4a) as the *EPAS1*'s downstream regulatory network. In total, 621 TGs are regulated by *EPAS1* via 1962 active REs (Supplementary Data 4). For example, EPAS1 binds to 13 REs to regulate transactivate vascular endothelial growth factor (VEGF), which promotes the growth of new blood vessels in high altitude adaptation. On average, each TG is regulated by 3.16 REs and the number of active REs follows a power-law distribution (Fig. 4b). Overall, EPAS1 tends to bind to REs within 300 kb to regulate downstream genes (Supplementary Fig. 9a).

We validated the *EPAS1* network by the significant overlap of 621 TGs with *EPAS1* knockdown by siRNA experiment in both HUVEC and C166[27] (p-values are $1.22 \times 10^{-4}$ and $7.8 \times 10^{-11}$, Fig. 4c, "Methods"). The EPAS1's target genes are also enriched in DEGs of lung and heart after heterozygous *EPAS1* knock-out in mouse[15] (*p*-value 0.018 and 0.01, hypergeometric test, Supplementary Fig. 9b, "Methods"). These 621 TGs tend to be positively selected in Tibetan and Andean populations (*p*-value 0.025 and 0.087, hypergeometric test, Supplementary Fig. 9c) and are differentially expressed in day 5 after hypoxia treatment (Supplementary Fig. 9d).

Functional enrichment of the 621 TGs reveals that angiogenesis and hypoxia response are two important terms (Fig. 4d). Then we present the core network for *EPAS1*'s regulation related

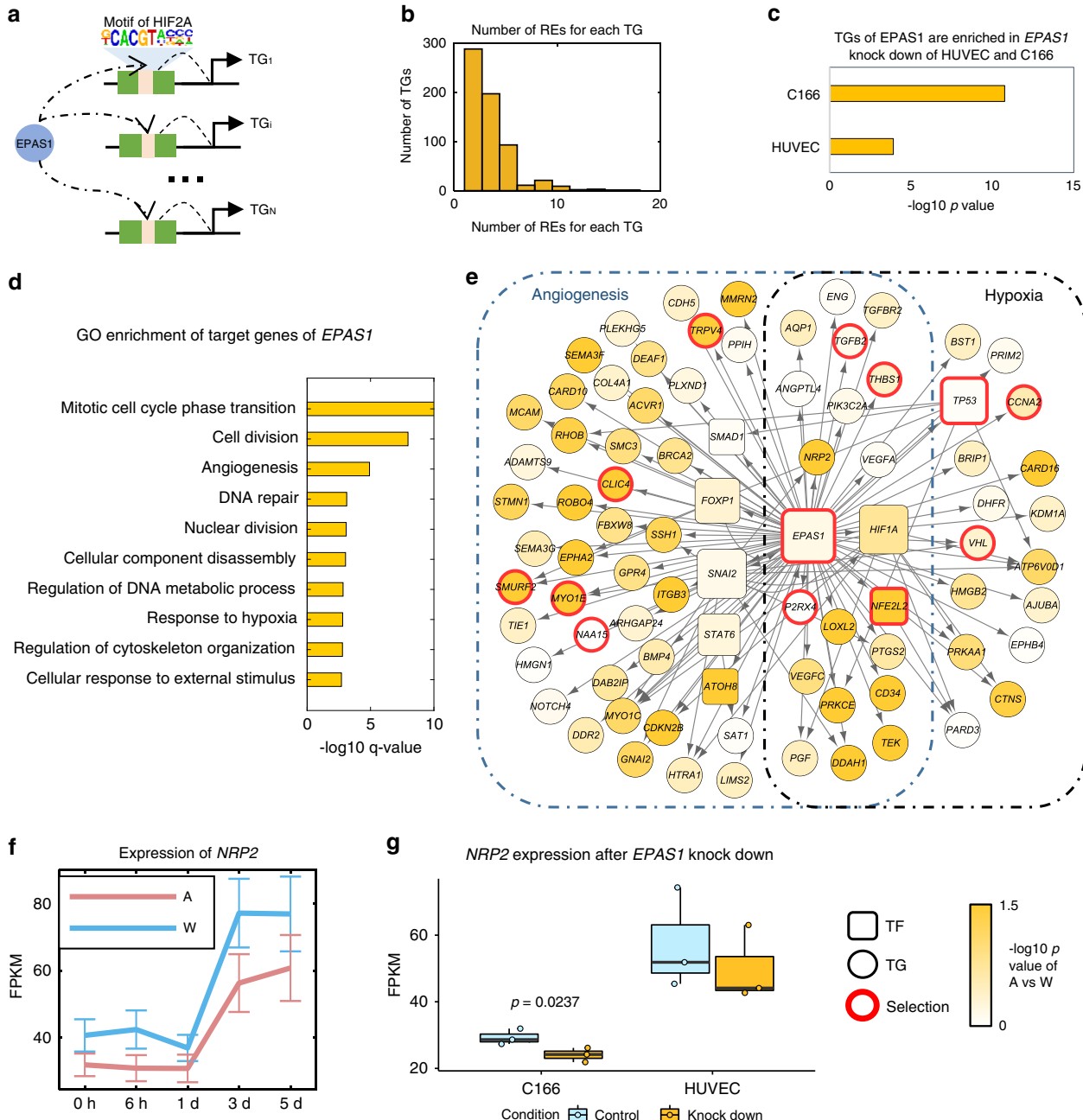

**Fig. 4 _EPAS1_'s downstream regulatory network. a** Extracting _EPAS1_'s subnetwork from the vPECA result. We extracted all the active REs with _EPAS1_'s motif binding link those REs with predicted TGs, and pool all the _EPAS1_-RE-TG triplets. In total, 621 TGs are regulated by _EPAS1_ via 1962 active REs. **b** The number of active REs for each TG follows a power-law distribution. **c** _EPAS1_'s 621 TGs are validated by the _EPAS1_'s RNAi knock-down experiment in HUVEC ($p = 1.22 \times 10^{-4}$, 56 overlap) and C166 ($p = 7.80 \times 10^{-11}$, 101 overlap) with hypergeometric test. **d** Functional enrichment of TGs reveals that angiogenesis and response to hypoxia are two important terms. _P_-values were calculated by hypergeometric test with Benjamini–Hochberg correction. **e** Core network for _EPAS1_'s regulation related to angiogenesis and response to hypoxia. Rectangles refer to TFs and circles are TGs. Genes with a red border are those with reported selection signals in Tibetan. Colors denote the $-\log_{10}(p\text{-value})$ of _t_-test of gene expression between A and W. **f** _EPAS1_'s target gene _NRP2_ shows blunted response to hypoxia at expression level. Data are presented as mean values ± standard error ($n = 5$). **g** _NRP2_ is differentially expressed after _EPAS1_ is knocked down in HUVEC and C166. _P_-value was calculated by one-sided _t_-test ($n = 3$). Boxplots are represented by minima, 25% quantile, median, 75% quantile, and maxima with data points. Source data are provided as a source data file.

to angiogenesis and hypoxia response (Fig. 4e), indicating that EPAS1 regulates hypoxia response TFs, including _FOXP1_, _SNAI2_, _HIF1A_, _ATOH8_, _NFE2L2_, _TP53_, as well as hypoxia-related genes with signals of positive selection in Tibetans. The regulation of _NOTCH4_ indicates the interplay between the cellular hypoxic response and the Notch signaling pathway[28]. Interestingly, EPAS1 regulates _NRP2_ by two distal REs (>600 Kb)

(Supplementary Fig. 9e), which is supported by the observed RE-promoter interactions in the Hi–C data (Supplementary Fig. 9f). _NRP2_ shows a blunted response to hypoxia at expression level (Fig. 4f) and the regulatory map shows a combination of active REs and active selected REs (Supplementary Fig. 9e). Moreover, independent data shows that _NRP2_ is differentially expressed after _EPAS1_'s knock-down in C166 (one-sided t-test,

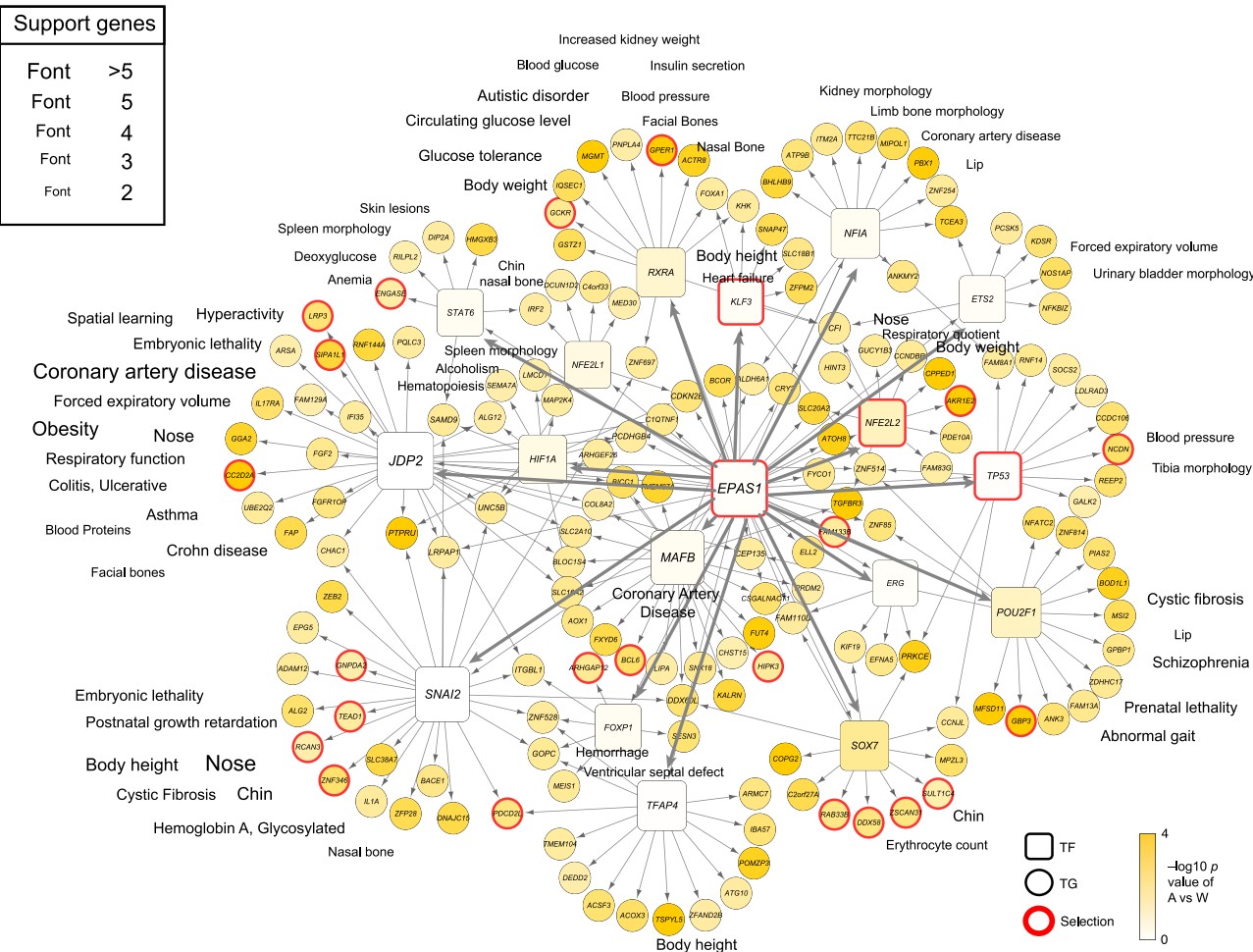

**Fig. 5 _EPAS1_-oriented regulatory network associates various phenotypes.** The _EPAS1_-oriented network is extracted from vPECA by selecting genes satisfying the conditions: the 1st and 2nd neighbor downstream of _EPAS1_, differentially expressed genes and enhancers between 0–6 h and 1–3 day (FDR < 0.05) and differentially expressed genes and enhancers between A and W (p-value < 0.05). Rectangles refer to TFs and circles are TGs. Genes with a red border are those with reported selection signals in Tibetan. Colors denote the $-\log_{10}(p$-value) of t-test of gene expression between A and W. Source data are provided as a source data file.

0.0237) (Fig. 4g). Literature search supports that NRP2 interacts with VEGF with convergent evolution in other highlanders. _NRP2-/-_ knockout mice display abnormal guidance and fasciculation of some cranial nerves and fewer small lymphatic vessels and capillaries[29].

**_EPAS1_-oriented network and its associations to the adaptive physiological traits in Tibetans.** To explore _EPAS1_'s regulation over the network, we extend the above network to an _EPAS1_-oriented subnetwork and associate it with phenotype data ("Methods" and Fig. 5). Our network provides the genetic basis and associations for diverse physiological traits and demonstrates the far-reaching role of _EPAS1_ as a major effect gene for adaptation. However, the network also implies the potential side effect for selecting a master regulator EPAS1, such as embryonic lethality, postnatal growth retardation, ventricular septal defect, and premature death (Fig. 5).

We also associate the network with the positively selected genes for high altitude adaptation in other species. _EPAS1_ was reported undergone positive selection in 13 species. _STAT6, HIF1A,_ and _FOXP1_ are positively selected in more than 2 species. We integrate the number of species showing positive selection into the 80-genes subnetwork (Supplementary Fig. 11). In total, there

are 80/178 genes (44%) showing evidence for positive selection in at least one organism other than human (Supplementary Data 5).

In addition to the _EPAS1_-oriented network, we also construct a hypoxia-oriented network by removing the _EPAS1_ neighbor downstream constraint and choose the TFs with dynamic expression and motif enrichment in active REs (Supplementary Fig. 10b). It has 208 genes and 440 interactions and is enriched with respiratory system development, circadian clock, vasculature development, cell cycle ($p$-value $< 10^{-5}$) and other terms (Supplementary Fig. 10c). This demonstrates _EPAS1_'s regulation propagation to multiple organs and processes via hypoxia response. _GPER1_ is far from the _EPAS1_ in the network with relatively weak selection (Fst > 0.15). It is differentially expressed between Tibetan and Han (two-sided t-test, $p$-value 0.01), and known to stabilize HIF-1a and promotes HIF-1a–induced VEGF and MMP9 in ESCs, which play critical roles in endometriosis[30,31] (Supplementary Fig. 10d).

**Genome-wide hypoxia and adaptation network provides a useful resource to interpret genetic variants.** We illustrate the use of the inferred active REs and regulatory network by vPECA to interpret the genetic variants with two examples.

Example 1: the vPECA network can annotate pulmonary hypertension (PAH) GWAS SNPs. PAH is a complex trait

without major-effect genes since no SNP achieved the statistical threshold for genome-wide significance ($P < 5 \times 10^{-8}$)[32]. We annotated the 319 risk SNPs with the most significant association p-values ranging from $1.82 \times 10^{-6}$ to $6.87 \times 10^{-4}$ after further association testing in an independent replication sample of 285 cases and 457 controls[32]. By overlapping the active REs in our network, we link the risk SNPs in REs to seven genes. In particular, *SOX17* is identified as a candidate gene for PAH. The risk SNP rs1995535 is located in the RE chr8:55246176-55252419, which is about 118 kb upstream to TSS of *SOX17*. vPECA links this risk SNP to *SOX17* rather than the nearest gene *LOC729038*. Another PAH associated locus rs10103692 (8:55258127: G/A), located 5 kb near our identified RE, is in an enhancer region that specifically regulates the expression of *SOX17* in endothelial cells[33]. Our analysis of GWAS data thus strongly implicates *SOX17* as a PAH associated gene. This is consistent with recent findings[34–36] that PAH patients are significantly enriched in rare deleterious variants. Also, vPECA predicts ERG as an upstream regulator of *SOX17* binding to the RE chr8:55246176-55252419 according to motif occurrence (Fig. 5), which is consistent with the fact that ERG binds to the super-enhancers of HUVEC and activates target genes including *SOX17*[17].

Example 2: The vPECA network can annotate Tibetans' structural variants (SVs). We identified 17,902 Tibetan-specific SVs from the long-read genome sequencing data ("Methods"). In addition to the SVs located in genes or introns, 638 SVs are found to overlap with the inferred active REs acting on distal target genes. Two SVs, a 200 bp duplication in chr9:139422831-139423031 and a 1200 bp duplication in chr9:139429296-139430493 are in the upstream RE of *NOTCH1* (chr9:139420177-139429697, 67,694 bp). They may increase this RE's accessibility, and affect *NOTCH1*'s expression, and may promote high altitude adaptation similar to the SNP rs3124608 located in this RE (Fst: 0.12)[28].

## Discussion

We provide an initial characterization of the chromatin accessibility and transcriptional landscape during hypoxia and adaptation. We develop a statistical methodology vPECA to fully utilize this dataset by integrating with public genomic data. Our integrative experimental and bioinformatics efforts provide a wealth of data resource, and data analysis method, and also a resource platform for interpreting genetic variants and biological insights. Our major contributions, which include a systematic approach for interpreting variants in the non-coding region, finding selected RE, model-based omics data integrating will have a broad interest in other fields. For population genetics study, we provide a genome-wide statistical method to detect causal regulatory element (selected REs) by paired expression and chromatin accessibility data for high altitude adaptation in Tibetans. Our multi-omics data integration on matched genome-transcriptome-DNA accessibility data is useful in precision medicine and personal omics[37]. For variant interpretation, our statistical approach is successful to annotate variants based on dynamical and conditional specific omics data. We propose that our vPECA model can be a general framework for GWAS variants interpretation ("Methods").

vPECA models the active selected REs, which significantly expands the traditional selected region concept (for example the 25 regions by DNA sequencing[11]), narrows down the regions to kb resolution, and reveals causal SNPs by RE's accessibility and the downstream target genes and upstream regulators. These REs can identify many candidate genes under selection, its regulation, and pathways. These genes are weakly selected in traditional population genetics studies but have a strong effect on

accessibility and expression in trait relevant cell types. We find many interesting genes in addition to *EPAS1*, especially those genes far from the selected REs. This also leads to the corresponding Tibetan vs Han difference regulatory subnetwork (Supplementary Data 2).

The integrative analysis of genetic variant, ATAC-seq, and RNA-seq by vPECA suggests a regulatory adaptation mechanism through selective fine-tuning among a set of active enhancers. The active REs are classified as either adapted or canonical depending on whether they show evidence of being selected. In the case of EPAS1, the canonical enhancer E12 is the major driver (accounting for 34% of the variance, Supplementary Fig. 6d) of the canonical hypoxia response in both Hans and Tibetans. In contrast, the adapted REs E21 and E22 exhibit selected changes in the Tibetan population. These changes fine-tune their accessibility dynamics to effect a blunted hypoxia response of EPAS1 expression (Supplementary Fig. 12a). We observed similar regulatory patterns in many other genes, i.e., *GCH1*, *NRP2*, *NQO1*, *NOTCH1*, *NOS3*, *HYOU1*, *BNIP3*, and *BCL6*, etc. (Supplementary Fig. 12b), which may imply a general regulatory mechanism for the blunted response. For example, *GCH1* and *NOS3* are involved in the blunted nitric oxide regulation in Tibetans under high-altitude hypoxia[38,39]. In the 1146 selected genes, 38 genes show significant expression differences between Tibetan and Han (p-value < 0.05). We check their regulatory map and most show the REs' combinatorial regulation. It will be interesting to further explore the canonical enhancers and adapted enhancers by public genomic data and features and its evolutionary evidence in other species.

vPECA reveals the feedback regulation of *EPAS1* and *HIF1A*, the most structurally similar and best-characterized genes in hypoxia. In mammals, the primary transcriptional response to hypoxic stress is mediated by these two hypoxia-inducible factors (HIFs)[40]. We observe HIF1a responses to hypoxia quite early at 6 h while *EPAS1* responses after 1d. *HIF1A* regulates *EPAS1* via active selected RE (E22 in Fig. 3e). In return, *EPAS1* negatively regulates *HIF1A*'s expression by four REs (Supplementary Fig. 13). This hypoxia stage-specific regulatory mechanism by two family members is consistent with prior studies indicating that *HIF1A* and *EPAS1* can promote the expression of distinct genes in endothelial cells[40].

Our current study takes advantage of *EPAS1/EGLN1* genotype biased design to reduce sample size and predicted enhancer target associations. The question remains for the extension from a single tissue to multiple tissues and to associate with the evolutionary time scale, i.e., how the variants related to complex traits are functional in different tissues? How evolution selects SNPs, elements, and genes in the hypoxia network to achieve high-altitude hypoxia adaptation accordingly along the evolution time in different species?

In the future, we will improve vPECA from several aspects. Hi–C without promoter capture identifies TAD boundaries but does not have the resolution to identify actual enhancer–promoter (E–P) interactions. For example, the Hi–C loop near *EPAS1* in Fig. 3a cannot identify E–P physical interactions. Nevertheless, Hi–C data are still useful to provide physical evidence for long-range interactions in *Sox17*, *SNAI2*, and *IL6* (Supplementary Fig. 15). The vPECA framework is ready to incorporate data from Trac-looping, HiCHiP, and CHIA-PET technologies, which will provide E-P interactions with higher resolution. We will further extend the vPECA model by considering self-regulation[41], solely using the available public data, providing a user-friendly interface, and integrating with other analyses in the eGPS framework[42]. We will continue our effort to provide biological interpretation for the signals of selection observed in the Tibetan population. For example, *TMEM247* is

another important gene near *EPAS1* with a strong signal of selection in Tibetans. *TMEM247*-rs116983452 has recently been identified to be significantly correlated with reduced hemoglobin concentration, red blood cell count, and hematocrit in Tibetans[43]. Because it is mainly expressed in testis, but not in HUVEC (Supplementary Fig. 16), we need to collect relevant tissues and generate multi-omics data to further explore its functional role. Finally, we will further improve our functional validation platform for the predicted causal SNPs, regulatory elements, and genes. For example, we plan to establish an immortal HUVEC cell line derived from Tibetans, which can be used in editing assays.

## Methods

**Modeling gene regulation with RE activity and selection**. The basic idea of the vPECA method is to model the distribution of expression of target genes (TG) and chromatin accessibility of regulatory elements (RE) conditional on expression of transcription factors (TF), chromatin accessibility of REs (O), selection status of SNPs on REs (X, Y), and linkage disequilibrium (LD) scores of those SNPs. vPECA uses four formulations to model, (1) expression of target genes, (2) activity status of the REs, (3) selection status of the REs, and (4) chromatin accessibility of REs (Fig. 2a and Table 1).

**TG expression**. We model a TG expression as a linear regression model shown in the following formulation.

$$TG_{i,j,l} = \beta_{l,0} + \sum_{k \in I_l} \beta_{l,k} Z_{i,j,k} \left( \sum_{m \in MB_k} \gamma_m B_{i,k,m} \cdot TF_{i,j,m} \right) + \varepsilon_l; \ \varepsilon_l \sim N(0, \sigma_l^2) \quad (1)$$

where $TG_{i,j,l}$ is the expression level of the $l$th TG of individual $i$ on time $j$. The error $\varepsilon_l$ is a Gaussian random variable with expectation zero and variance $\sigma_l^2$. We assume that the $l$th TG expression level is determined by the activation status of its regulating REs denoting as $I_l$ and the TF complex binding to these REs. The strength of TF complex binding to $k$th RE of individual $i$ on time $j$ is denoted as $\sum_{m \in MB_k} \gamma_m B_{i,k,m} \cdot TF_{i,j,m}$, where $B_{i,k,m}$ is the sum of all matching strength of $m$th TF binding to $k$th RE of individual $i$. To simplify the complexity of motif binding sites when considering individual specific sequence, we only use motifs scanned from the reference genome. The expression of $m$th TF of individual $i$ on time $j$ is denoted as $TF_{i,j,m}$. Moreover, $MB_k$ represents the motif binding sites on the $k$th RE. $\beta$, $\gamma$, $\sigma^2$ are the parameters to be estimated.

**RE activity**. We assume that RE activity is determined by both genomic and epigenomic features, i.e., RE activity is modeled by linear regression of selection signals and open chromatin dynamics represented by ATAC-seq signals, which is revealed by the following formulation.

$$Z_{i,j,k} = O_{i,j,k} + \alpha_k \delta_i S_k + \varepsilon_{zk}; \ \varepsilon_{zk} \sim N(0, \sigma_{zk}^2), \quad (2)$$

where hidden variable $Z_{i,j,k}$ denotes the activation status of the $k$th RE of the $i$th individual on time $j$. We assume that each RE's activity status Z follows a normal distribution, where the mean is modeled by the combination of $O_{i,j,k}$ and $S_k$. Here, $O_{i,j,k}$ is the openness score of the $k$th RE of individual $i$ on time point $j$, while $S_k$ denotes the selection signals of the $k$th RE defined by formulation (4). The variance is denoted by $\sigma_{zk}^2$. $\delta_i$ is an indicator function to represent population information.

$$\delta_i = \begin{cases} 1, & \text{if individual } i \text{ is Tibetan} \\ 0, & \text{otherwise} \end{cases}. \quad (3)$$

$\alpha_k (k = 1, \dots, K)$ are parameters to be estimated, while $K$ is the total number of REs.

**Selection signals of RE**. When modeling the difference of gene regulation between Tibetans and populations living at low altitude, REs under positive selection are under great consideration. We defined variable $S_k (k = 1, \dots, K)$ as the selection status of the $k$th RE. $S_k$ is measured by several widely used test scores for selection signals denoted as $X_{p,q} (p \in J_k, q = 1, \dots, Q)$, which represents the $q$th selection test score of the $p$th SNP on the $k$th RE, where $J_k$ denotes the set containing all selection SNPs on the $k$th RE, while $|J_k|$ is the total number of SNPs containing in $J_k$. Q equals to the total number of selection test scores (Here, $Q = 4$). In our case, $X_{p,q} (q = 1, \dots, 4)$ refer to the $-\log_{10}$ p-value of Fst, iHS, XP-EHH, and PBS respectively. $Y_p (p \in J_k)$ represents the ΔDAF score of the $p$th SNP on the $k$th RE ("Methods"—"Calculating selection scores for SNPs" section). The selection status of RE is modeled by a logistic regression corresponding to the weighted summation of selection test scores X and Y.

$$\log \frac{P(S_k = 1 | X, Y)}{1 - P(S_k = 1 | X, Y)} = \mu_0 + \sum_{q=1}^{Q} \mu_q \sum_{p \in J_k} \frac{1}{|J_k|} w_p X_{p,q} + \mu_{Q+1} \sum_{p \in J_k} \frac{1}{|J_k|} w_p Y_p, \quad (4)$$

where $w_p$ indicates the weight of the $p$th SNP, which is related to LD score, defined by formulation (5). $\mu$ is the parameters to be estimated.

One challenge in population genetics is to consider the LD among SNPs. Here we introduce the weight of each SNP by utilizing the LD related network. We represent SNPs as a network format (Supplementary Fig. 8e), where a node denotes a SNP and edge denotes LD score after thresholding. If two SNPs have a correlation larger than a given threshold, then there exists an edge between them. The degree of a fixed node in the network represents the number of SNPs linkage to the fixed SNP. Mathematically, let $r_p$ be the LD score of SNP p, the weight of $p$th SNP $w_p$ is defined as the reciprocal of LD score of SNP p, i.e.,

$$w_p = 1/r_p. \quad (5)$$

Here, $w_p$ is used to down-weight the influence of SNPs with large LD, which reduces the redundancy of correlated SNPs. LD score based on the population of East Asian from 1000 Genomes project is collected from the URL (https://data.broadinstitute.org/alkesgroup/LDSCORE/).

**Chromatin accessibility of RE**. We model chromatin accessibility of a RE with a linear formulation, i.e.,

$$O_{i,j,k} = \eta_{jk} + \omega_{jk} \delta_i S_k + \varepsilon_{ok}; \ \varepsilon_{ok} \sim N(0, \sigma_{ok}^2), \quad (6)$$

where $O_{i,j,k}$ is the openness score of the $k$th RE of individual $i$ on time point $j$, while $S_k$ denotes the selection signals of the $k$th RE defined by formulation (4), and $\delta_i$ is a population indicator defined by formulation (3). We assume openness score of the $k$th RE at time $j$ follows a normal distribution with mean $\eta_{jk} + \omega_{jk} \delta_i S_k$, and variance $\sigma_{ok}^2$. $\eta_{jk}$ is the average openness level of the $k$th RE at time $j$ in both populations, while $\omega_{jk}$ is a parameter to balance $\eta_{jk}$ and $S_k$. Both $\eta$ and $\omega$ are parameters to be estimated. The assumption is that the openness score of a RE can be partially determined by the DNA sequence, i.e., selection status $S_k$. With $\omega_{jk} \neq 0$, we extract those REs under selection and the chromatin accessibility in time $j$ are different between populations.

**Likelihood function**. The likelihood function of our statistical model is as follows.

$$P(TG, O | TF, O, X, Y) = \sum_{Z,S} P(TG, O, Z, S | TF, O, X, Y)$$
$$= \sum_{Z,S} P(TG | TF, Z) P(Z | S, O) P(O | S) P(S | X, Y),$$

where TG and TF indicate the observed expression data of target gene and TF, and O represents observed chromatin accessibility data of regulatory elements. X and Y are the selection test scores of each SNP derived from DNA sequence data. In this model, we aim to maximize the likelihood function to estimate parameters $\mu$, $\alpha$, $\beta$, $\gamma$, $\omega$, $\sigma^2$, as well as hidden variables Z and S.

$$\max_{\mu,\alpha,\beta,\gamma,\omega,\sigma^2} P(TG, O | TF, O, X, Y; \mu, \alpha, \beta, \gamma, \eta, \omega, \sigma^2). \quad (7)$$

This is equal to

$$\max_{\mu,\alpha,\beta,\gamma,\eta,\omega,\sigma^2} \sum_{Z,S} P(TG | TF, Z; \beta, \gamma, \sigma_l^2) P(Z | S, O; \alpha, \sigma_{zk}^2) P(O | S; \eta, \omega, \sigma_{ok}^2) P(S | X, Y; \mu)$$

where $P(TG | TF, Z; \beta, \gamma, \sigma_l^2)$, $P(Z | S, O; \alpha, \sigma_{zk}^2)$, $P(S | X, Y; \mu)$ and $P(O | S; \eta, \omega, \sigma_{ok}^2)$ are computed according to Eqs. (1), (2), (4) and (6).

**Estimate $P(S|X, Y; \mu)$ by iteratively updating $\mu$ and S**. Note that the selection status of each RE is mainly determined by population genetics information, i.e., it is independent of chromatin accessibility and gene expression, the selection status Z (Eq. (4)) is estimated independently from other equations. For estimation problem of $P(S|X, Y; \mu)$ denoted by Eq. (4), we implemented the following steps.

(1) Let $p_k^0 = 1 - \frac{1}{|J_k|} \sum_{p \in J_k} c_p$, where $c_p$ denotes the p-value of CMS score of the $p$th SNP. Initiate $P(S_k = 1) = p_k^0$ and

(2) Estimate $\mu$ given $P(S_k = 1 | X, Y)(k = 1, \dots, K)$ by least-squares:

$$\min_{\mu} \sum_k \left\| \log \frac{P(S_k = 1 | X, Y)}{1 - P(S_k = 1 | X, Y)} - \mu_0 - \sum_{q=1}^{Q} \mu_q \sum_{p \in J_k} \frac{1}{|J_k|} w_p X_{p,q} - \mu_{Q+1} \sum_{p \in J_k} \frac{1}{|J_k|} w_p Y_p \right\|_2^2.$$

(3) Estimate $P(S_k = 1 | X, Y)$ given $\mu$:

$$P(S_k = 1 | X, Y) = f \left( \mu_0 + \sum_{q=1}^{Q} \mu_q \sum_{p \in J_k} \frac{1}{|J_k|} w_p X_{p,q} + \mu_{Q+1} \sum_{p \in J_k} \frac{1}{|J_k|} w_p Y_p \right),$$

where f denotes the sigmoid function, i.e., $f(x) = 1/(1 + e^{-x})$.

(4) If not convergent, go to step (3). Convergence means $\mu$ does not change between adjacent steps.

After the above iteration, we obtained the selection status $P(S_k = 1 | X, Y)$ for all REs. We choose a threshold of 0.95. If $P(S_k = 1 | X, Y) > 0.95$, $P(S_k = 1 | X, Y) = 1$. Otherwise $P(S_k = 1 | X, Y) = 0$. This procedure enforces sparsity to the REs under selection.

**Estimate $P(O|S;\eta,\omega,\sigma_{ok}^2)$.** Equation (6) is a linear regression model. given $O$ and $S$, parameters $\eta$, $\omega$, and $\sigma_{ok}^2$ can be easily estimated by the least square. Then we used t-distribution to test whether $\omega$ is non-zero, with $p$-value $< 0.05$.

**Estimate $P(TG|TF,Z;\beta,\gamma,\sigma_l^2)$ and $P(Z|S,O;\alpha,\sigma_{zk}^2)$.** Combining Eqs. (4) and (6), we obtained Eq. (8).

$$TG_{i,j,l} = \beta_{l,0} + \sum_{k \in I_l} \beta_{l,k}(O_{i,j,k} + \alpha_k \delta_i S_k)\left(\sum_{m \in MB_k} \gamma_m B_{i,k,m} \cdot TF_{i,j,m}\right) + \varepsilon_l. \quad (8)$$

Then we updated $\alpha$, $\beta$, and $\gamma$ iteratively by fixing two parameters and calculating the other one. Specifically, for each TG, we solved the following optimization problem.

$$\min_{\alpha,\beta,\gamma} \sum_{i,j} \left\| TG_{i,j,l} - \beta_{l,0} - \sum_{k \in I_l} \beta_{l,k}\left(O_{i,j,k} + \alpha_k \delta_i S_k\right)\left(\sum_{m \in MB_k} \gamma_m B_{i,k,m} \cdot TF_{i,j,m}\right) \right\|_2^2. \quad (9)$$

Note that usually, the numbers of REs and TFs are much larger compared with the number of samples, we implemented variable selection method LASSO when estimating $\beta$ and $\gamma$. Moreover, in order to make good use of regulatory relationships in public data, we calculate prior information $\beta_0$ (correlation between RE and TG) and $\gamma_0$ (correlation between TG and TF) for $\beta$ and $\gamma$. Based on public data of paired chromatin accessibility (DNase-seq and ATAC-seq) and gene expression data across ~200 cellular contexts, we calculate the cross-tissue correlation between gene expression and openness score of RE of all candidate RE-TG pairs and obtained $\beta_0$, and also co-expression between all TGs and TFs, which forms $\gamma_0$.

For each TG, we implement the following algorithm:

(1) Initiate $\gamma = \gamma_0$ and $\alpha = 0$, solving Lasso:

$$\min_\beta \sum_{i,j} \left\| TG_{i,j,l} - \beta_{l,0} - \sum_{k \in I_l} \beta_{l,k} O_{i,j,k}\left(\sum_{m \in MB_k} \gamma_m^0 B_{i,k,m} \cdot TF_{i,j,m}\right) \right\|_2^2 + \lambda\|\beta\|_1,$$

where $\gamma_m^0$ indicates the prior of the mth TF with the lth TG. We used 5-fold cross-validation to choose $\lambda$. Then REs with $\beta \neq 0$ are chosen to the next step.

(2) Fix $\beta$ calculated from step (1) and $\alpha = 0$, solving the following LASSO problem:

$$\min_\gamma \sum_{i,j} \left\| TG_{i,j,l} - \beta_{l,0} - \sum_{k \in I_l} \beta_{l,k} O_{i,j,k}\left(\sum_{m \in MB_k} \gamma_m B_{i,k,m} \cdot TF_{i,j,m}\right) \right\|_2^2 + \lambda\|\gamma\|_1.$$

Similarly, we use 5-fold cross-validation to choose $\lambda$. Then TFs with $\gamma \neq 0$ are chosen to the next step.

(3) For TFs and REs chosen from step (1) and (2), we calculate the following non-linear optimization model.

$$\min_{\alpha,\beta,\gamma} \sum_{i,j} \frac{1}{2} \left\| TG_{i,j,l} - \beta_{l,0} - \sum_{k \in I_l} \beta_{l,k}\left(O_{i,j,k} + \alpha_k \delta_i S_k\right)\right.$$

$$\left.\left(\sum_{m \in MB_k} \gamma_m B_{i,k,m} \cdot TF_{i,j,m}\right) \right\|_2^2 - \beta_0^T \beta - \gamma_0^T \gamma + \frac{1}{2}\|\beta\|_2^2 + \frac{1}{2}\|\gamma\|_2^2.$$

We used the "fminunc" function in MATLAB to solve the above problem, where the quasi-newton method is implemented in the nonlinear programming solver. Thus, $\alpha$, $\beta$, and $\gamma$ are obtained.

**Output of vPECA.** vPECA identifies 32,330 active REs (ARE) ($\beta \neq 0$) for 9952 genes including 1,647 active selected REs (ASRE) ($\Pr(S = 1) > 0.95$, $\alpha \neq 0$ and $\beta \neq 0$) for 1146 genes (Supplementary Data 2) and the associated regulatory network including 52,647 interactions among REs and target genes. In addition, we use $\omega \neq 0$ to select those selected REs with differential accessibility in local time points. vPECA also identifies TFs potentially binding to REs with $\gamma \neq 0$. Thus we obtain 428,871 TF-RE-TG triplets in total. If we omit the REs, 109,090 TF-TG relations are identified.

**Genome annotation in HUVEC.** We note that a good genome annotation is available that contains the coordinates of all transcriptional units (genes) and most REs in the genome[44]. A RE is defined as a short region in the chromosome, typically a few hundred bp in size, on which sequence-specific TFs and other related proteins may assemble to exert control on the transcription of nearby genes. ENCODE has mapped more than 68,636 REs by H3K27ac and H3K4me1 ChIP-seq in the HUVEC cells. In addition, high-resolution 3D chromatin interactions from Hi–C data in HUVEC[45] impose physical boundaries for promoter-enhancer interactions. Population genetics studies have identified thousands of positively

selected variants underlying high-altitude adaptation with various quantitative scores and their LD associations (Fig. 2a).

**Motif scan and identification of enriched dynamic TFs.** In the bubble plot of Fig. 1d, we select a set of TFs based on their expression level (FPKM) and motif enrichment scores. We use Homer[46] to scan motifs on peak regions of each ATAC-seq sample. The motif enrichment (ME) score is defined as

$$\text{ME Score} = \sqrt{-\log_{10}P \times \text{FoldChange}}.$$

At each time point, we have 10 samples (5 Hans and 5 Tibetans). The color of each dot represents the average gene expression value and the circle size indicates the average ME score along with all samples at each time point. As Fig. 1d shows, the gene expression FPKM values are divided into 5 levels, i.e. Level I: $< 6$, Level II: 6–12, Level III: 12–30, Level IV: 30–120, and Level V: $> 120$, which are represented by 5 colors. Similarly, the ME scores are divided into 7 levels showing by the circle size. Then we filter out motifs by two requirements, i.e. (1) Maximal FPKM $\geq 12$, and (2) Maximal ME Score $\geq 2$, which gives 43 TFs (long list in Supplementary Fig. 3a). Moreover, we require that the TFs should be dynamic, i.e. the expression FPKM and ME scores are not at the same level along 5 time points, thus obtaining 14 TFs in the short list (Fig. 1d).

**Calculating selection scores for SNPs.** For each SNP, the selection signals are represented by several commonly used test scores based on public sequenced population genome data. Since the population data with a large number of samples could provide rich information on human genetics. In this study, we obtained 4,627,029 variants in total after variant calling from the whole-genome sequencing data of 38 Tibetan highlanders and 39 Han Chinese lowlanders[8]. Then for each SNP, we compute the fixation index (Fst)[47], iHS[48], XP-EHH[49], and PBS[14] scores. Each score is represented by its $-\log10$ $p$-value. To composite multiple signals, we use Fisher's method to combine all the 4 scores together and calculated an overall $p$-value on chi-squared distribution, then FDR $< 0.05$ is implemented and 111,182 SNPs are identified as under selection. This method is a simplified version of the widely used CMS score[50].

Additionally, we incorporate delta derived allele frequency (ΔDAF) of the 10 WGS data to capture the difference between 5 Tibetans and 5 Han individuals and to reveal the specific DNA sequence information of our hypoxia induction experiment. ΔDAF is defined as the derived allele frequency of 5 Tibetans minus the derived allele frequency of 5 Hans.

**Enrichment of SNPs under selection in functional regions.** The first critical step for our vPECA model is to quantify the aggregation effect of variants in non-coding regions. The aim is to choose the right cell type and condition for GWAS variants interpretation and then generate multi-omics data in this condition. This step is extremely important when regulatory elements are functional in dynamic contexts. We calculate fold enrichment (FE score of selection SNPs in functional genomic regions. For each SNP set under a certain cutoff (for example, 111,182 SNPs are under selection with FDR $p < 0.05$), we calculate their fold enrichment score defined by the following formulation.

$$\text{FE score} = \frac{\#(\text{SNPs above the threshold in peaks})/\#(\text{AllSNPs in peaks})}{\#(\text{total SNP above the threshold})/\#(\text{total SNPs})},$$

Where total SNP number equals to 4,627,029. Then FE scores are calculated across 40 DNase-seq tissues from ROADMAP (Supplementary Fig. 4c) and 50 ATAC-seq samples in our hypoxia induction experiment. For all 50 samples, we calculate the average FE score along 5 samples under each condition (within one population in one-time point) (Supplementary Fig. 4b), where each line represents the FE score of one population in one particular time point under changes of selection cutoff. Then if the FE scores are averaged by time points (each time contains 10 samples, ignoring population difference), we obtain the fold enrichment changes under normoxia (0 h) and hypoxia (6 h) (Fig. 1f).

**Approach for fold enrichment of SNPs in functional regions.** For 111,182 SNPs under selection (FDR $< 0.05$), we calculated their fold enrichment scores in annotated regulatory regions such as enhancers and promoters in ENCODE. For each functional region set, we calculated the fold ratio by the formulation:

$$\text{Fold ratio} = \frac{\#(\text{Selection SNPs in regions})/\text{region length}}{\#(\text{total SNP})/\text{whole genome length}},$$

where the whole-genome length is 3 billion base pairs and the number of total SNPs is 4,627,029 (Supplementary Fig. 4a). $P$-value is calculated using binomial distribution with distribution function b(x,n,p), where $n = 111,182$, $p = $ Region length/whole-genome length, and $x = \#$(Selection SNPs in regions).

With the same method, we also calculated the enrichment score of 31 high Fst SNPs around EPAS1 in active enhancer regions, i.e., H3K27ac peaks, of multiple tissues in ROADMAP (Fig. 3i, j and Supplementary Fig. 8f).

Nine genes with Pearson correlation coefficient (PCC) larger than 0.5 are listed in Supplementary Fig. 8f, which providing candidate selection genes that may have functions in a variety of cellular contexts other than HUVEC. Among the genes, *LDHA* expression is mediated by HIF1/2 A in human pancreatic cancer[51]. And

LDH family is under selection in adaptation to hypoxia in yaks[52]. This gene is highly expressed in A549 lung, K562 leukemia cells, and other epithelial cells. *NEK7* was identified under selection in Tibetan according to another publication[9]. And it's highly expressed in the aorta, fibroblast cells, and right atrium.

**Criteria for choosing the donors**. Written informed consent was obtained from each subject. We report sample collection and primary culture of human umbilical vein endothelial cells (HUVECs) as follows[15]. Umbilical cords were obtained from 131 normal, full-term pregnancies at the People's Hospital of Lhasa, Tibetan Autonomous Region of China. Written informed consent was obtained from each subject. We implemented stringent criteria, deep surveys, as well as genetic analyses in the sampling procedure to make sure that the ancestry of the donors are Tibetans.

Firstly, the ancestry of the donors was determined by the self-report and verified by the genetic analyses using genotyping and whole-genome sequencing data. Only those Tibetans, whose lineal relatives are also Tibetan ethnicity within three generations, were included in this study. The same strategy was also implemented for Han donors.

For genetic verification, we chose the adaptive (Tibetan) and wildtype (Han Chinese) populations according to the *EPAS1/EGLN1* genotypes using Tibetan-specific common SNPs (Supplementary Data 11). Given the DAF (derived allele frequency) of these SNPs in the Han population is extremely low (near 0), we can determine ancestry with high confidence using their genotypes. Additionally, we used known ancestry populations conducted principal component analysis, which also supports the ancestry identification from genotypes.

For pregnancy situation, we confirmed that the Tibetan donors were all born and raised at high altitude (Lhasa, elevation: 3680 m), and their entire pregnancy procedure took place at high altitude and no low altitude visit during the period. Biologically related individuals were excluded from this study.

**Choosing relevant cell type to hypoxic response**. We chose HUVEC to study hypoxia response by the following reasons.

(1) HUVEC can be easily accessed compared with other tissues, such as lung and heart in human. Hypoxia adaptation is a complex physical trait that involves multiple tissues in the respiratory and circulatory systems. These tissues are important when interpreting selection signals but difficult to access. Moreover, although gene regulatory networks are different between tissues, hypoxic inducible pathways are conserved to some extent. The HIF family genes, such as *HIF1A* and *EPAS1*, have similar expression patterns responsive to hypoxia.

(2) HUVEC is a classic model to study oxidative stress and hypoxia, due to its oxygen-sensitive and easy to obtain. We found many references using it to study cellular responses to hypoxia[53–56]. Nakato et al. recently cataloged gene expression and active histone marks in nine types of human ECs (generating 148 genome-wide datasets) and carried out a comprehensive analysis with chromatin interaction data. They pointed out that endothelial cells (ECs), which make up the innermost blood vessel lining of the body, express diverse phenotypes that affect their morphology, physiological function and gene expression patterns in response to the extracellular environment, including the oxygen concentration, blood pressure and physiological stress[57].

(3) HUVEC is listed as the Tier 2 ENCODE Project common cell types [https://www.genome.gov/encode-project-common-cell-types]. Cell types were selected largely for practical reasons, including their wide availability, the ability to grow them easily, and their capacity to produce sufficient numbers of cells for use in all technologies being used by ENCODE investigators. Importantly, HUVEC has a normal karyotype and are readily expandable to $10^8$–$10^9$ cells. ENCODE and NCBI GEO database collected rich public data for HUVEC, i.e., RNA-seq, DNase-seq, and ChIP-seq experiments, which provides much independent validation samples.

(4) *EPAS1* (endothelial PAS domain protein 1) is a key transcription factor involved in hypoxic induction and Tibetan adaptation to high-altitude. Compared with *HIF1A*, *EPAS1* expression is more tissue restricted[58] and highly expressed in endothelial cells, such as HUVEC, which makes it an appropriate cell type in this study.

**Generating multi-omics data and data quality control**. We design experiments to collect multi-omics data, in particular high-quality ATAC-seq data for chromatin accessibility landscape, to interpret the positively selected variants underlying high-altitude adaptation. To improve our ability in identifying the causal target genes with reasonable sample size, we measure functional genomics data from 10 individuals with different genetic backgrounds (5 Tibetans, 5 Han Chinese with desired *EPAS1/EGLN1* genotypes, "Methods") during time-course hypoxia with matched genome sequence, transcriptome, and chromatin accessibility data. To meet the inherent cell type specificity challenges, HUVEC is chosen as the convenient and oxygen-sensitive human endothelial cells from the large vein of the umbilical cord, which has rich public data available in ENCODE. Figure 1a shows our procedures for adaptive and wildtype population choosing, individual filtering

by *EPAS1/EGLN1* genotypes, HUVEC cell selection, time series hypoxia induction, and multi-level omics data profiling (details in "Methods").

We collected high-quality RNA-seq and ATAC-seq data for 50 samples (HUVEC from 10 donors, each with 5 time points) to assess gene expression and chromatin accessibility as a hallmark of active DNA regulatory elements (Supplementary Data 1). For ATAC-seq data, each sample has more than 100 M uniquely mapped reads (Supplementary Fig. 1a), two-fold enrichment in transcription start site (TSS) (Supplementary Fig. 1b), and clear fragment length distribution to show nucleosome structure (Supplementary Fig. 1c). All the 50 samples showed significant consistency with HUVEC than other cell types in the ENCODE data (Supplementary Fig. 1d). Moreover, we collected whole-genome sequencing data of the 10 cell lines and obtained Hi–C data of HUVECs of 2 Tibetans and 2 Han Chinese from both 0 h and 3 day (8 samples in total) for mapping 3D chromatin interactions. Data processing and quality checking details are described in "Methods". Our rich data provide a foundation to assess the regulatory landscape of high-altitude adaptation.

**Controlling the cell culture effect in the data**. To evaluate the effects of cell culture in normoxia, we performed additional experiments that cells are cultured in normoxia for the same lengths of time ("Methods"). We generated RNA-seq data and analyze them together with the hypoxia data. We found that the variance of samples being cultured in hypoxia (75%) is much larger than in normoxia (11%) by ~7 fold (Supplementary Fig. 2a). And the number of DEGs in hypoxia response is significantly larger than in normoxia by more than 30 folds (1000 vs 30 in Supplementary Fig. 2b). DEGs between 0 h and 5 day in normoxia are not related to response to hypoxia (Supplementary Fig. 2c). Thus, we confirm that response to being in culture for different periods of time are not significant compared to the responses to hypoxia.

**Quantifying positively selected variants**. Variants are called from whole-genome sequencing data by distinguishing 38 Tibetan highlanders and 39 Han Chinese lowlanders[8]. We quantify the selection by combining Fst, iHS, XP-EHH, and PBS scores ("Methods") and identified 111,182 variants (among the 4,627,029 tested variants) under selection (FDR < 0.05). They are enriched in the annotated regulatory regions such as enhancers and promoters in ENCODE (Supplementary Fig. 4a). Furthermore, those variants are enriched in the open regions of our 50 HUVEC samples with varying thresholds.

**Endothelial cell hypoxic assay**. The HUVECs were isolated from the veins of human umbilical cord by the well-established method[59]. Briefly, cords were separated from placentas within 2 h after delivery. The umbilical vein was cannulated and washed through with sterile phosphate buffered saline (PBS), supplemented with 100 units/ml penicillin, 0.1 mg/ml streptomycin, and 5.6 mg/ml amphotericin B in order to remove any blood clots. It was then infused with 1 mg/ml (125 U/ml) of type I collagenase (Gibco) in Ca2 and Mg2 free PBS and incubated at 37 °C for 15 min. After incubation, the HUVECs were flushed from the vein vessel together with the collagenase solution. The obtained HUVECs were cultured on flasks coated with 0.1% gelatin. The cells were grown in complete medium (Medium199 supplemented with 10% FBS, 20 mmol/l HEPES (pH 7.4), 1 ng/ml recombinant human fibroblast growth factor (bFGF), 1 ng/ml endothelial growth factors (EGF), and 100 units/ml penicillin, 0.1 mg/ml streptomycin) (Gibco) at 37 C in 5% CO2, and the medium was changed every 2–3 days. About 7 to 10 days, the primary culture cells were confluent in the flask. For subculture, cells were detached and harvested with a 0.05% trypsin-EDTA solution, then neutralized with FBS (Gibco). When the cells were cultured to passage-3, they were cryopreserved with cell freezing medium (NCS + 10% DMSO) in liquid nitrogen. The protocol of this study was approved by the Institutional Review Board of the Kunming Institute of Zoology, Chinese Academy of Sciences.

All hypoxia experiments were performed in passage-5 cells. The HUVECs were divided into two groups adaptive vs. non-adaptive based on the genotypes of *EPAS1* (including TED) and *EGLN1* (Fig. 1a), and each group had five cell lines. Cells were placed in a conventional 37 °C humidified incubator (Thermo-Forma, Model 3141) with automated gas control of 1% O2, 5% CO2, balanced with N2. The oxygen concentration of the hypoxic incubator was constantly monitored by a real-time O2 sensor with 0.1% sensitivity. Here the choice of 1% O2 is empirical and mainly based on the literature[60–63]. Our choice of 1% O2 is roughly estimated based on the following understandings. Oxygen partial pressures of cells and bodies are different. Oxygen is inhaled by the body to reach the mitochondria of the cells, and the partial pressure of oxygen is a progressive step down. At sea level (normoxia), oxygen partial pressure is 159.22 mm hg (20.95%), but the partial arterial oxygen pressure (PaO2) is only 100 mmHg (13%), while the venous oxygen partial pressure (PvO2) is 40 mmHg (5.2%), and the mitochondrial oxygen partial pressure is 4–20 mmHg (0.52–2.6%). Each cell line was divided into four dishes for whole-genome sequencing, RNA-seq, ATAC-seq, and Hi–C in parallel, and harvested at different time points, including 0 h, 6 h, 24 h (1 day), 72 h (3 days) and 120 h (5 days). We selected all 50 samples for RNA-seq and ATAC-seq libraries. All of 10 cell lines were extracted DNA for whole-genome sequencing and 8 samples (2 cell lines each group at 0 h and 72 h) for Hi–C libraries.

**RNA-seq library preparation and data processing**. Total RNA was isolated using TRIzol reagent (Ambion). Libraries from the resulting total RNA were prepared using the TruSeq paired-end mRNASeq kit. Then library preparations were sequenced on an Illumina Hiseq 4000 platform and 150 bp paired-end reads were generated.

We used TopHat[64] to map sequencing reads to human reference genome hg19. Then used Cufflinks[65] to call FPKM values as gene expression measurement. Genes with FPKM value 0 in all samples were filtered out. Genes that are not expressed consistently within replicates, i.e., the ratio of maximum to minimum FPKM among 5 replicates is larger than 500, were also filtered out. Finally, 12,998 protein-coding genes remain for further study.

**ATAC-seq library, sequencing, and data preprocessing**. We performed ATAC-seq as described[66]. We used the TruePrep DNA Library Prep Kit V2 for Illumina (Vazyme, TD501) to prepare the ready-to-use DNA library for ATAC-Seq. followed the Instruction manual, fifty thousand cells (no fixation) were centrifuged at 500×g for 5 min at 4 °C. Washed once with 50 μL of cold 1x PBS buffer. Centrifuge at 500×g for 5 minutes at 4 °C. Gently pipet to resuspend the cell pellet in 50 μL of cold lysis buffer (10 mM Tris-HCl, pH = 7.4, 10 mM NaCl, 3 mM MgCl 2, 0.1% IGEPAL CA-630). Immediately centrifuge at 500×g for 4 min at 4 °C. Discard the supernatant and add 50 μL transposition MIX to resuspend the cell pellet, then placed in PCR machine for the transposition reaction at 55 °C 10 min. After purified the transposition product with AMPure® XP beads, proceed PCR Enrichment immediately for 9 cycles. Purify the amplified library AMPure® XP beads for size selection. Real-time PCR and Qubit® were used to definitely quantify the concentration of the library. ATAC-seq libraries were sequenced on Illumina Hiseq 4000 platform and 150 bp paired-end reads were generated.

ATAC-seq pair-end reads were trimmed for Illumina adaptor sequences and transposase sequences with a customized script and then mapped the reads to reference human genome hg19 with Bowtie v1.0.0[67]. Duplicate reads were removed with Samtools.v0.1.19[68]. Only uniquely aligned reads were used for peak calling with Hotspot using default parameters. We calculated a QC score as the ratio of total read counts at TSS over a 2-kb window to the randomly selected background for each sample.

**Whole-genome DNA sequencing and variant calling**. The genomic DNA was extracted by HUVEC lysis using the standard phenol-chloroform method and sequenced by Illumina X10 sequencer with 150 bp paired-end reads. We generated the average 124 Gb raw sequence data for per subject (Supplementary Data 6). We performed the quality control by FastQC and Genome Analysis Toolkit (GATK), the mean Q30 of read-pairs is higher than 86%. Each sample has the reads depth >36X (Supplementary Data 6).

We used the standard GATK pipeline to call the variants. With GRCh37 (hg19) as the reference genome, we mapped reads to reference by bwa-MEM module[69], then Picard was utilized to mask the PCR duplicates and produced the dedup.bam. Local alignment was performed to adjust the alignments via GATK indel realignment, GATK score recalibration modules. Then, we used the known site set as training data, re-calculated base quality via base quality score recalibration (BQSR) module. Finally, variants were called by GATK HaplotypeCaller module[70] and made the genomic VCF (gVCF) file, then we performed join calling of all gVCF via GenotypeGVCFs module and obtain the raw VCF file as candidate variant set. For variant filtering, the variant quality score recalibration (VQSR) module of GATK was used based on the arguments listed in Supplementary Data 7.

**Hi–C library, sequencing, and data preprocessing**. Hi–C library was performed as described in ref. [71]. One million Cells were fixed by final concentration 1% formaldehyde, mixed for 10 min at room temperature. Fixation was quenched using 2.5 M glycine at room temperature for 5 min and immediately centrifuged at 800×g for 10 min. The supernatant was removed, the pellet resuspended in lysis buffer (final concentration of 10 mMTris, pH 8.0, 10 mM NaCl, 0.2% Igepal CA-630 and 1x complete protease inhibitors (Sigma-Aldrich) and incubated on ice for 10 min. Removal of lysis buffer was done by centrifugation at 2500×g for 5 min at 4 °C, followed by the removal of the supernatant. The pellet was resuspended in 342 μl 1x NEBuffer 3.1 and incubated with 38 μl 10% SDS at 65 °C for 10 min. Next, add 43 μl of 10% Triton X-100 to the Hi–C-tube to quench the SDS, 37 °C for 15 min. Add 12 μl of 10x NEBuffer 3.1 and 400U DpnII, mix gently Digest the chromatin overnight at 37 °C on a rocking platform. On the next day, the DpnII restriction enzyme was inactivated at 65 °C for 25 min. Next, fill-in Biotin by adding biotin-14-dATP, dCTP, dGTP, dTTP, and DNA polymerase I Klenow, incubate at 23 °C for 4 h. The digested chromatin was diluted and religated in by T4 DNA ligase, incubated at 16 °C for 4 h, and shaken manually 3 times. The chromatin products were de-cross-linked overnight by adding 30 μl proteinase K, and incubated at 65 °C overnight. Extract the DNA, dissolved in 50 μl 10 mMTris, pH 8.0. Next, using T4 DNA polymerase to remove biotin from un-ligation ends, 20 °C for 4 h, inactivate the enzyme for 20 mins at 75 °C. Shear the DNA to a size of 200–300 bp using Covaris M220. Using AMpure XP magnetic beads perform DNA size selection. Biotin pulldown with Streptavidin T1 beads. End repair, add A, add adaptor reaction and PCR amplification, DNA product size selection was

performed step by step. The Hi–C library was sequenced with Illumina Hiseq 4000 platform for paired-end sequencing.

Totally we have eight samples, two replicates of both populations in 0 h and 5 days. We used two methods to process and analyze Hi–C data. First, HiC-Pro was used to process Hi–C data from raw sequencing reads to normalized contact matrix[72]. The main steps include mapping raw reads to hg19 reference genome, detecting valid ligation products, and generating raw contact maps. For raw contact matrix generating, we set a variety of resolutions of 500 bp, 1 kb, 5 kb, 10 kb, 25 kb, 50 kb, 100 kb, 500k, and 1 Mb, which means that genome is divided into bins with the above equal sizes. Then we used the iterative correction method[73] to eliminate the systematic basis and generate the normalized contact matrix. According to the publication that "map resolution" is defined as the smallest locus size such that 80% of loci have at least 1000 contacts[45]. If we merge the two replicates of the same population in each time point, i.e. W–0 h, A–0 h, W–3 day, and A–3 day, we will gain 5-kb resolution of four samples. In the second step, we used Homer[46] to further identify significant interactions (FDR < 0.01) based on contact maps generated by HiC-Pro under 5 k resolution.

The proportion of duplicate reads and different contact ranges are shown in each Hi–C sample. For each sample, we removed duplicated reads, and then divided remaining valid reads into cis long-range (>200 k), cis short-range (<200 k), and trans contacts, which are demonstrated by different colors. The proportion of duplicate reads and contact ranges in eight original samples and four merged samples are shown in Supplementary Fig. 14. The figure demonstrates that our experiment is in high quality since the fraction of long-range intra-chromosomal valid pairs for each sample is significantly larger than 40 %[72]. We obtained 48–64% for 8 samples, and 50–61% for 4 merged samples.

**Luciferase reporter assay**. We chose E2, E12, E20 (rs370299814, rs368706892), E21 (rs569774785), E21 (rs4953357, rs6756667), E22 (rs10206434, rs141366568), and E22 (rs3768729) to verified by luciferase reporter assay[74]. The 100-bp synthetic single-strand oligonucleotides were annealed to form double strands consisting of the corresponding genotypes (CC and TT) for rs569774785, (GG, AA and AA, GG) for rs4953357 and rs6756667, "AA, GG and GG, AA" for rs10206434 and rs141366568, "AA, TT and GG, CC" for rs370299814 and rs368706892 and "T and C" for rs3768729 flanked by restriction sites. The fragments were cloned into the multiple cloning site of the pGL3-promoter vector (Promega). All constructs of new-building plasmids were validated by sequencing to make sure no de novo mutation was introduced. The presumably adaptive alleles are "C" for rs569774785, "G, A" for rs4953357, rs6756667;"A, G" for rs10206434, rs141366568;"A, T" for rs370299814, rs368706892 and "T" for rs3768729 (Supplementary Fig. 8a). The reporter vectors containing either an adaptive allele or no adaptive allele were co-transfected into HEK293T and HeLa cells respectively, together with a reference vector (pRL-TK vector). HeLa and HEK293T cells were grown in Gibco Dulbecco's Modified Eagle's Medium (Gibco) supplemented with 10% fetal bovine serum (HyClone). Lipofectamine 2000 (Invitrogen) was used in transient transfection. After 36 h incubation in 21% oxygen (normoxia) and 1% oxygen (hypoxia) conditions, we collected the cell lysates and measured luciferase activity using the Dual-Luciferase Reporter Assay System (Promega, Madison, WI). The relative light units were measured using a luminometer. The mean values of three independent experiments were used.

**HUVEC cultured in normoxic and hypoxic condition**. We collected 2 replicates of HUVEC from one Han Chinese and cultured them in normoxia (21% oxygen) for five days. Other cell culture conditions are controlled the same as the hypoxic induction experiments. RNA-seq was performed on 0 h, 6 h, 1 day, 3 day, and 5 day for both replicates. The raw data were processed following the same pipelines in the RNA-seq data processing part. The total reads numbers and the alignment statistics of RNA-seq data are shown in Supplementary Data 1B, indicating it is in high quality. Next, we compared the responses to being in culture for different periods of time to the responses to hypoxia. We pooled the data together, carefully removed the batch effect by ComBat[75], and checked the variations of gene expression data between cells being cultured under normoxia and hypoxia. We used PCA to visualize the two datasets after batch effect adjustment (Supplementary Fig. 2a). The normoxic samples (solid circles) are close to the previous 0 h samples (red samples). The variance between normoxia samples (solid circles) is relatively smaller than the hypoxia ones (hollow shapes), especially in 3 day and 5 day. In the PCA plot, the largest variance revealed by PC1 is the response to hypoxia, i.e., the difference between normoxia to hypoxia 1 day and 3–5 days under hypoxia (PC1 75%). This is the same as the results in Fig. 1c. PC2 shows the difference between samples cultured in normoxia and hypoxia along time and is a mixture of response to culture time and hypoxia response. There are two key observations from the plot. (1) the responses to being in culture at normoxia for different periods of time are insignificant and negligible compared to the responses to hypoxia. Although cells are different between 0 h and 5 day when cultured under normoxia, the variance is much smaller than 3–5 day under hypoxia (PC1 75% vs PC2 11%). Thus, the effect that cells respond to hypoxia is much larger than responses to being in culture in normoxia for different periods of time. (2) the response to culture along time is quite independent, dramatically different, and not related to the response to hypoxia in PC2. They have opposite directions that originated from the very similar 0 h samples. This indicates there are two different gene groups

underlying response to hypoxia and response to culture time. To further dissect the general patterns revealed by PCA, we compared the differentially expressed genes (fold change of the average value of two groups larger than 2) between different time periods under normoxia and hypoxia. The numbers of DEGs between adjacent time points in hypoxia response, especially 1 day vs 3 day and 3 day vs 5 day, are significantly larger than in normoxia by more than 30 folds (the number of DEGs are 1000 vs 30 in Supplementary Fig. 2b). In total, we identified 560 DEGs between 0 h and 5 days in normoxia with fold change >2. The functional terms are significantly enriched in cell cycle and DNA replication (Supplementary Fig. 2c, p-value < $10^{-40}$), which are not related to hypoxia response.

We used HEK293 and HELA to do dual-fluorescent reporting experiments to test whether the elements are functional as enhancers. The transfection of the dual-fluorescent reporting system is usually transient, so the dual-fluorescent reporting system often uses cell lines of high transfection efficiency, easy culture and rapid growth such as HEK293 and HELA. The transfection efficiency of the primary culture HUVEC is <10%, and our current culture system only allows the generation of the HUVEC for 92 h. Therefore, the signals could not be detected by the dual-fluorescence reporting system.

**Public data collection.** Totally 40 DNase-seq, 98 H3K27ac, HUVEC H3K4me1, H3K4me3 peak files (bed file), and 57 gene expression datasets were collected from Roadmap Epigenomics Project (ROADMAP). This includes 40 cellular contexts contain matched H3K27ac and gene expression data. We also downloaded eQTL data across 44 tissues from GTEx Portal (The GTEx Analysis V6p release was downloaded from GTEx Portal on 11/14/2017)[76], which was used to provide the genetic level evidence for the RE-TG pairs predicted by vPECA. Whole genome sequencing data of 38 Tibetan highlanders and 39 Han Chinese lowlanders[8] were used to compute population genetic selection scores. H3K27ac histone marks under normoxia (0 h) and hypoxia (24 h) are also downloaded from[21]. In addition, we collect high-resolution 3D chromatin interactions from HiC data in HUVEC[45] to impose physical boundaries for promoter-enhancer interactions.

**RE annotation and openness score quantification.** Totally 40 DNase-seq, 98 H3K27ac, HUVEC H3K4me1, and H3K4me3 peak files (bed file) were collected from Roadmap Epigenomics Project (ROADMAP)[77] and Encyclopedia of DNA Elements (Encode) database[78]. Hi-C loops of HUVEC measured 3D chromatin interactions were downloaded from[45]. For each gene, we defined the regulatory boundary as the [−1 Mbp, +1 Mbp] region to TSS that overlaps with the nearest loop contains the gene. If the loop is larger than 1 M to TSS, the boundary is just [−1 M, +1 M], otherwise, it is the overlap region within the loop. Then within the regulatory boundary of each gene, we defined candidate REs as the functional regions, with both H3K27ac and H3K4me1 marks (overlap with peak files of these two marks), as well as promoters [−2 k bp, 0] to TSS. Now we have a candidate list of REs that may potentially regulate certain genes based on the distance to TSS and Hi-C loops. Then we calculated openness score for each RE under each condition as the fold change of reads number per base pair[19]. For all 81,634 REs defined by either the overlap of two histone marks or promoters, we calculated their openness scores under 50 conditions. The REs with openness score 0 along with all samples and inconsistent within replicates were filtered out (same methods with gene expression, refer to "RNA-seq library preparation and data processing" in Supplement methods). Finally, 51,406 REs remains for further study.

**Hierarchical clustering and principal component analysis.** For the expression data of 12,998 genes and chromatin accessibility of 51,406 REs under 50 conditions (contains 5 time points of two populations), we performed hierarchical clustering and PCA. For hierarchical clustering, we implemented correlation as clustering distance and "complete" as a clustering method. The heatmap was plotted with R package pheatmap[79]. The first two principal components are shown in Fig. 1c.

**Differential analysis of genes and REs.** For the differential analysis of both gene expression and chromatin accessibility data, we used limma to perform design matrix[80]. For each sample, we implemented a binary variable "Population" to indicate whether the sample belongs to Han Chinese or Tibetan, and variable "Time" with 5 categories (0 h/6 h/1 day/3 day/5 day) to represent different time points. We also incorporated variable "individual" to indicate the 10 individuals between 5 time points. Then for each gene or RE, we constructed the following linear model:

$$y = \sim 1 + Population + Time + individual,$$

where y indicates a vector of expression of a certain gene or openness score of a RE. With this linear model, we designed various contrast matrix for differential analysis between populations and adjacent time points. Then we controlled FDR < 0.05 to identify differentially expressed genes and REs. Specifically, when calculating DEGs between every pair of adjacent time points, we used 0 h–1 day, 1–3 day, 3–5 day for both W and A separately (Fig. 1e). For DEG analysis between A and W in each time point, we used W.time- A.time (0 h/6 h/1 day/3 day/5 day) as the contrast matrix, and p-value < 0.05 as a threshold.

**Functional enrichment analysis of DEG and DOR in dynamics.** We used DAVID[81] to analyze the functional enrichment of DEGs. The number of DEGs between every two adjacent time points are listed in Supplementary Data 8. For each population, we did functional enrichment analysis separately. In general, the function terms of DEGs are very similar between Tibetans and Han Chinese (Supplementary Data 9). In comparison between 0 h and 6 h, glycolytic process and carbo metabolism are important both in A and W. From 6 h to 1d, other than glycolytic process, mitochondrial function and DNA damage response genes are differentially expressed. When comparing 1d with 3d, because the number of DEGs is too large to use in DAVID, we divided the DEGs into two groups, i.e. Time1 > Time2 and Time1 < Time2. Enriched terms include mitochondrial functions, DNA repair, canonical glycolysis, and VEGF signaling pathway. In all, 3–5 day also contains canonical glycolysis as a key factor.

For differentially accessed regions, we used GREAT[82] to annotate their functions where default parameters were set. We listed functions in the MSigDB pathway of those DCAs between adjacent time points. Also, two populations share similar functions. Key enriched functions include HIF-1/2-alpha transcription factor network, IL6-mediated signaling events, VEGF and VEGFR signaling network, and Actions of Nitric Oxide in the Heart, which are consistent with prior knowledge of hypoxia induction process and functions of DEGs above.

**RE-promoter interactions revealed by reads count of Hi-C.** To show the potential physical interactions between RE-promoter identified by vPECA model, we used the Hi-C data from four samples (W–0 h, A–0 h, W–3 day, and A–3 day). We fixed the target gene's promoter region and extracted the raw read counts within every 1k bin (including REs) from the raw contact matrix (1 k) into a sparse matrix. Each promoter may contain several bins, we combined these bins into a single promoter region to get the interaction reads between the promoter and all the 1k bins and REs. Since four samples have different sequencing depths, we normalized the raw reads count by dividing the total reads counts in the sample. Then, we drew the scatter plot of read counts supporting bin-promoter interactions for each bin along with genome positions. To visualize the potential interactions, we smoothed those data points using locally weighted scatterplot smoothing (LOESS). Those peaks in the curve are consistent with the predicted RE-promoter interactions (Fig. 3d, Supplementary Fig. 9f). To make sure the trend by the LOESS method is stable, we tried different parameter settings for a smoother span. Furthermore, we used Bootstrap to resample the data points 300 times and plot the average of LOESS curves. We observed a similar trend. In addition, we tried different bin sizes from 1 k, 1.5 k, 2 k, to 4 k, the peaks shown in RE-promoter interactions are consistent with predicted results. Similarly, we can also focus on one particular RE rather than a promoter, and find interactions between this RE region with other bins (Supplementary Fig. 7).

**Validation of 621 target genes of EPAS1 by siRNA knockdown.** The cell lines of HUVEC and C166 were transfected with EPAS1 siRNA and non-targeting control siRNA for 48 h. The efficiencies of knocked down the EPAS1 expression were assessed by qPCR with 1.4% for HUVEC and 3.2% for C166[27]. The FPKM gene expression data of HUVEC and C166 was downloaded from the GEO database with accession number GSE62974. Then EPAS1's target genes were derived by comparing expression profiles of EPAS1 siRNAs with non-targeting ones. We calculated the fold change (FC) of average FPKM values between replicates and identified 795 genes in HUVEC and 1382 in C166 with FC larger than 1.5. These genes were assumed to be true positive to validate 621 EPAS1 target genes derived from the regulatory network. Then we used the hypergeometric test to calculate the enrichment of vPECA predicted target genes of EPAS1 in the gold standard gene set.

**Enrichment analysis with hypergeometric test.** We used the hypergeometric test to calculate the enrichment of predicted active REs in H3K27ac peaks under hypoxia in 24 h (Fig. 2b) and predicted selection genes enriched in literature supported genes (Fig. 2c). Also, we used the hypergeometric test to compute the enrichment of TGs of EPAS1 in DEGs in EPAS1 knockdown in mouse (Supplementary Fig. 9b) and selection genes in other populations (Supplementary Fig. 9c) and DEGs along time (Supplementary Fig. 9d). For enrichment analysis of EPAS1 target genes, the hypergeometric cumulative distribution function is given by

$$F(x|K, M, N) = \sum_{i=0}^{x} \frac{\binom{K}{i}\binom{M-K}{N-i}}{\binom{M}{N}},$$

where x is the number of predicted target genes of EPAS1 in the gold-standard gene set. M denotes the total number of expressed genes in HUVEC. K indicates the number of gold standard genes expressed in HUVEC. N is the number of predicted EPAS1 target genes.

**Functional enrichment of multiple gene sets.** We used Metascape[83] to do the gene functional enrichment analysis. Metascape selected significantly enriched

function terms and clustered similar terms together (Figs. 2e and 4d, and Supplementary Figs. 5b, and 10a, d).

We annotated the gene functions of the *EPAS1* oriented network (Fig. 5) using the MGI (Mouse Genome Informatics) database [http://www.informatics.jax.org/] and GWAS Central [https://www.gwascentral.org/], which involves the phenotypes of gene-knockout mouse and significantly associative traits/diseases in human cohorts. Functions were marked when the supported gene number is larger than two.

**Subnetwork extraction**. For network analysis, vPECA provided TF-RE-TG triplets that reveal the whole picture of the transcription regulation network ("Methods"—"Output of vPECA" section).

For core subnetwork of *EPAS1*'s downstream analysis (Fig. 4e), we focused on all *EPAS1*-TG pairs, which means fix TF as *EPAS1* and select 621 downstream target genes. Among 621 genes and EPAS1, we selected genes related to "response to hypoxia" and "angiogenesis", then constructed the core network in Fig. 4e. Edges denote all predicted TF-TG relations between these genes.

EPAS1 oriented network (Fig. 5) mainly considers the 1st and 2nd neighbor downstream genes of *EPAS1*. For all TF-RE-TG triplets within the 2nd order of the *EPAS1* downstream network, we selected TGs and REs under the following two conditions. (1). Differentially expressed genes and REs between 0–6 h and 1–3 day (FDR < 0.05). (2). Differentially expressed genes between A and W (*p*-value < 0.05). Here, differential genes and REs are calculated from "Differentially expressed genes (DEG) and differential chromatin accessibility analysis" in "Methods". The network encompasses 178 genes (18 TFs) and 234 interactions totally.

Hypoxia-oriented network (Supplementary Fig. 10b) mainly considers the hypoxia-responsive network related to adaptation. For all TF-RE-TG triplets in the whole network computed by vPECA, we extracted a subnetwork under two conditions. (1). Differentially expressed genes and REs between 0h-6h and 1–3 day (FDR < 0.05). (2). Differentially expressed genes and REs between A and W (*p*-value < 0.05). Then from the above subnetwork, we selected the top 25 enriched TFs by defining an enrichment score, i.e. the number of TFs occurred in the subnetwork divided by the number of TFs in the whole genome. The number of TF occurrences is based on motif binding sites. Usually, one TF matches several motifs, so the TF occurrence number in the background is defined as the maximum occurrence number among all motifs match to the same TF.

**EPAS1-oriented network construction and annotation**. To explore *EPAS1*'s regulation over the network, we extend the above network to an *EPAS1*-oriented subnetwork and associate it with phenotype data (Fig. 5). The network mainly considers the 1st and 2nd neighbor downstream genes of *EPAS1* under the two conditions. (1) Differentially expressed genes and REs between 0–6 h and 1–3 day (FDR < 0.05); (2) Differentially expressed genes and REs between Tibetan and Han (*p*-value < 0.05). The network has 178 genes (18 TFs) and 234 interactions with 17 modules, which is enriched in oxidative stress and developmental terms (Supplementary Fig. 10a). The modules were annotated with MGI (Mouse Genome Informatics) phenotypes[84] and GWAS Central database[85] ("Methods"). Many adaptive physiological traits in Tibetans show associations in the network such as blood pressure, erythrocyte count, forced expiratory volume, respiratory quotient, spleen/kidney/limb bone morphology, and coronary artery disease, etc. These are consistent with the Tibetan's advantage stemming from a different hypoxia regulation of ventilation, a low hypoxia pulmonary vasoconstrictor response, better lung function, higher maximum cardiac output, better levels of blood oxygen saturation, and better sleep quality[6]. As expected, heart and lung related physiologies and diseases are the most enriched items in these modules (14 modules are involved). Coronary artery disease (CAD), heart failure, respiratory functions (e.g. forced expiratory volume (FEV)) and a series functions about blood index, e.g. erythrocyte count, blood pressure, hemoglobin, blood glucose and anemia are consistent with the existing study for the main functions of *EPAS1* in Tibetans ("Methods").

**Phenotype annotation for *EPAS1* oriented subnetwork**. We constructed the *EPAS1* regulation network according to the putative interactions ("Methods"—"Subnetwork extraction" section). *EPAS1* regulated 197 downstream genes, including 17 TF as a mediator and forming 17 subnetworks. To explore the potential function for these subnetworks, we performed functional enrichment using MGI (Mouse Genome Informatics) database and the GWAS Central database.

As expected, heart and lung related physiologies and diseases were the most enriched items in these subnetworks (14 subnetworks were involved), for example, coronary artery disease (CAD), heart failure, respiratory functions (e.g. forced expiratory volume (FEV)) and a series functions about blood index, e.g. erythrocyte count, blood pressure, hemoglobin, blood glucose and anemia, which was consisting with the previous study for *EPAS1*'s major functions in Tibetans[15,56,86–90].

Notably, we observed some intriguing functions were enriched in some subnetworks (Fig. 5). For example, functions related to facial structures were enriched in eight subnetworks, including nose features (nasal bones), chin, and

lips. It is reported that nasal variation is related to the high-altitude adaptation in Tibetans and Andeans, which may serve in increased oxygen uptake and air-conditioning processes in highlanders[30]. Besides, body mass indexes (weight and height) were enriched in five subnetworks, although it is still lacking a systematic epidemiological survey for comparison of stature situation in Tibetans and lowlanders. Some reports propose that Tibetan adults and adolescents possess a shorter stature in high altitude than lowlanders[91,92]. In addition, other functional traits were observed in multi-subnetworks, including spleen morphology, skin lesions, and glucose tolerance (Fig. 5). These enriched functions were reported before as an adaptive trait or a compensatory action. For example, selection has increased Bajau (a kind of breath-hold diving people) spleen size, which provides an oxygen reservoir for hypoxia diving[93], but it needs more data to identify the role of spleen morphology change in Tibetan high-altitude adaptation. Besides, it is reported that Tibetan highlanders were vulnerable to glucose intolerance[94], but it is hard to tell whether the phenotype contributes to high-altitude adaptation. Of course, the functional enrichment of these network provides insights for Tibetan hypoxia adaptation, for example, we can speculate that Tibetans may exhibit some adaptive characters for protecting skin from high UV in a high-altitude environment (no reports) based on our function enrichment in the subnetwork of STAT6 (Fig. 5). In summary, we generated a systematic *EPAS1*-downstream network-based multi-omics data and summarized the potential functions for *EPAS1* downstream genes, which may be putative candidate genes for adaption of high altitude in Tibetans.

**Annotate Tibetans' structural variants by the vPECA network**. Structure variations (SVs) may play important roles in human adaption to extreme environments such as high altitude. For example, a 3.4-kb copy number deletion near *EPAS1*, named TED, is significantly enriched in high-altitude Tibetans. Many lines of evidence support that the TED is a promising candidate that might have played a critical role in high altitude adaptation of Tibetans[95]. Here, we combined long-read sequencing with next-generation mapping and assembled a high-quality Tibetan genome (ZF1), with a contig N50 length of 24.57 mega-base pair (Mb) and a scaffold N50 length of 58.80 Mb. We detected 17,902 SVs, and 6,505 of them are ZF1-specific when compared with two previously published de novo lowlander Asian genomes (AK1 and HX1).

The whole-genome long read and short read sequencing data are available at The Genome Sequence Archive (GSA) [http://gsa.big.ac.cn/index.jsp] under the project ID of PRJCA000936.

**vPECA framework for systematic GWAS variants interpretation**. Our study also has implications for human genome-wide association studies (GWAS), where more than 90% of disease-associated regions do not affect proteins directly, but instead lie in non-coding regions with putative gene-regulatory roles. We show that our vPECA model can be applied to GWAS variants interpretation in the following aspects: (1) Quantify the aggregation effect of variants in non-coding regions; (2) Generate multi-omics data in the right cell type and condition. This is extremely important when regulatory elements are functional in dynamic contexts. For example, active selected REs are functional under hypoxia pressure and ultra-conserved enhancers are functional in developmental process[96]; (3) Integrate multi-omics data and reconstruct regulatory network. We expect understanding the mechanism of genetic variants will go beyond the current candidate variants from GWAS study and static profiling by ENCODE annotation. We propose a feasible and efficient strategy as vPECA annotation (Variant plus time course paired expression and accessibility data) in the relevant tissues. We see the potential that vPECA can be generalized to understand the complex traits such as schizophrenia with the prefrontal cortex's multi-omics data and face phenotype with neural crest cells' multi-omics data.

**Reporting summary**. Further information on research design is available in the Nature Research Reporting Summary linked to this article.

## Data availability

The data generated in this study, including RNA-seq, ATAC-seq, and Hi–C data were deposited at [http://www.ncbi.nlm.nih.gov/geo/] (accession number: GSE145774) and Genome Sequence Achieve (project number: CRA002025; [https://bigd.big.ac.cn/gsa/browse/CRA002025]). All other relevant data supporting the key findings of this study are available within the article and its Supplementary Information files or from the corresponding authors upon reasonable request. Source data are provided in Zip folder. Source data are provided with this paper.

## Code availability

The source codes for vPECA are available at [https://github.com/AMSSwanglab/vPECA].

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

## Acknowledgements

We would like to thank all volunteers who provide samples for this study. The authors also thank Prof. Bianbazhuoma (People's Hospital of Lhasa City) and Drs. Yi Peng, Xiaoming Zhang for providing helps during sampling, and Prof. Luonan Chen, Peng Shi, and Rui Jiang and other lab members for helpful discussions. This work was supported jointly and equally by grants from the National Natural Science Foundation of China (NSFC) (91631306 to B.S.) and the Strategic Priority Research Program of the Chinese Academy of Sciences (CAS) (XDA20040102 to X.Q.; XDB13000000 to B.S. and Y.W.). Additional resources were provided by grants from the NSFC (11871463, 61671444, and 61621003 to Y.W., 31660308 to Ou.), the Youth Innovation Promotion Association of CAS (to Y.H.), the State Key Laboratory of Genetic Resources and Evolution (GREKF18-07 to C.B.), Shanghai Municipal Science and Technology Major Project (2017SHZDZX01 to Y.W.), and the Natural Science Foundation of Yunnan Province (2019FB042 to H.Z.). W.H.W. and Z.D. were supported by NIH grants P50HG007735, R01GM109836, and R01HG010359.

## Author contributions

B.S., Y.W., and W.H.W. conceived and supervised the project. J.X., Y.H., and Z.D. designed the experimental/analytical approach and performed numerical experiments and data analysis. H.Z., X.L., and C.C. performed all biological experiments. L.C., C.Z., O. Z., C.B., and X.Q. contributed analysis or biological materials. J.X., H.Z., Y.H., Z.D., C.B., L.C., X.L., D.Y., C.Z., X.Z., Q.Y., Z.F., C.C., X.Q., O.Z., W.H.W., Y.W., and B.S. wrote, revised, and contributed to the final paper.

## Competing interests

The authors declare no competing interests.
