## [Peer Review File · Nature Communications]

Reviewers' comments:

Reviewer #1 (Remarks to the Author):

This manuscript presents a valiant effort to provide a biological interpretation for the signals of selection observed in Tibetan populations, particularly as they apply to the EPAS1 locus. To that end, they gather new data on transcript levels and chromatin dynamics in Tibetan and Han HUVECs exposed to hypoxia for increasing time lengths. Some of this data is then used in conjunction with an extension (vPECA) of a previously developed method to interpret the likely targets of selected regulatory elements in the Tibetan genome. This article presents a wealth of new data and data analysis that the authors rightly point out is key to better understanding and interpreting non-coding variation in human adaptation. However, this manuscript seems to fall apart in its attempt to apply their analytical approach to the EPAS1 locus. Several major concerns exist with regard to their experimental set up, the application of the method and the functional validation of regulatory elements. This article may hold promise as a resource or methods development paper, but it should not be suitable for publication in Nature Communications as written.

Major concerns:

- 1.) The experimental set up for the RNAseq and ATACseq data is based on culturing cells in hypoxic conditions for increasing lengths of time including the 0h time point. Transcriptional or chromatin accessibility responses are then measured relative to different time points. This is not the standard set up for hypoxia response experiments where cells are cultured in parallel in normoxia and hypoxia for the same lengths of time. The experimental design used in this manuscript cannot distinguish between responses to hypoxia and responses to being in culture for different periods of time, which greatly complicates the interpretation of results.
- 2.) With regard to the Hi-C data, it is well established that Hi-C without promoter capture identifies TAD boundaries, but does not have the resolution to identify actual enhancers. While the Hi-C data generated by the authors is valuable in its own right and may be useful in the PECA modeling, it cannot be used as a validation of specific enhancers, as presented – for example – in Fig. 3A. Incidentally, this figure is misleading because they do not represent other genes, besides EPAS1, that are also present within the same TAD (eg PRKCE). Also, it is unclear why in some analyses they use Hi-C data from previous publications and in others they use their own data.
- 3.) No information is provided with regard to subjects that donated the umbilical blood cords other than the hospital and their self-reported ethnicity. Were they all born and raised at high altitude? Did the entire pregnancy take place at high altitude? Did they visit low altitude locations during the pregnancy or during the past 6 months? Transcript levels and chromatin conformation are molecular phenotypes that can be modified by environmental exposure especially in the context of hypoxic conditions. Was the ancestry of the donors verified by genetic analyses?
- 4.) While the authors focus on a single SNP (rs3768729) and region (E22-1) which appear to show evidence of reduced enhancer activity and chromatin openness in the Tibetan-enriched haplotype, they fail to discuss other regions on the same tightly linked haplotype which show the exact inverse patterns in their analysis. For example, the SNPs in E20 consistently show the inverse effect (the wildtype allele significantly decreases enhancer activity and is associated with reduced chromatin openness). Within E22, the two regions/SNPs tested appear to go in opposite directions with regards to the adaptive alleles' association with chromatin openness. While these may not entirely invalidate the overall conclusions regarding multiple enhancers contributing to the downregulation of EPAS1 in Tibetans, they need to be addressed rather than buried in the supplement.
- 5.) rs3768729 which the authors describe as having a striking effect on enhancer activity, and putatively EPAS1 expression in Tibetans based on their reporter gene assays is a common SNP worldwide. The putatively adaptive allele is ancestral and segregates at .48-.58 frequency globally. If this SNP had such a profound effect on enhancer activity and EPAS1 expression, it is surprising that it has not been detected in any GWAS or eQTL study given this high frequency. The authors should address this point if they wish to put this SNP forward as the most probable and/or important causal SNP under selection at the EPAS1 locus.

- 6.) The choice of cell type, while interesting, is not well justified. The authors claim the endothelium is hypoxia-responsive and therefore a good choice of tissue in which to assess Tibetan chromatin structure. This statement seems to assume either that the findings within the endothelium are representative of other hypoxia-responsive tissue types (which is unlikely, given that regulatory networks will vary between tissues) or that endothelium is the most relevant tissue type in which to examine hypoxia adaptation. If the latter, the authors need to do a better job of explaining this choice.
- 7.) As far as I can tell, the reporter gene assays were performed in normoxic conditions where both HIF1A and HIF2A are tagged for ubiquitination and degraded. Therefore, these are not the appropriate conditions for validating the enhancers and SNP effects and the results highly questionable.
- 8.) To assess the significance of the composite multiple signals the authors combine p-values using Fisher's approach (lines 742-745), but this approach assumes that the p values from independent tests of the same hypothesis, which is clearly not the case here because SNPs are in LD thus not independent.
- 9.) The project accession PRJNA544821 is not found in Bioproject.

Minor Comments:

- 1.) The putatively selected regulatory regions defined by vPECA are typically much larger (>1kb) than the regions looked at for differential openness (~200-550bp) which are again larger than the very short segments examined in the luciferase assays (100bp).
- 2.) The cell types chosen for the luciferase assays are not endothelial and enhancer activity in these contexts may not be representative of the endothelial regulatory network.
- 3.) In lines 360-361 the authors describe RORA as having "the second strongest signal for high-altitude adaptation" in Ethiopian populations. However, the articles they cite for this claim explicitly state that while a SNP in RORA is the second strongest signal of association with Hb concentration in the Amhara, it has no signal of selection in their data. Therefore, the statement above is false or at least misleading.
- 4.) The paper as written is littered with typos that would need to be fixed in a final edit.
- 5.) While the Authors present a compelling data analysis pipeline, they appear to apply this model specifically to their Tibetan expression and chromatin accessibility data, while using publicly available data for the other aspects of the model. For the purposes of method development, it would be interesting to determine how similar the results would be if, for instance, ENCODE data on HUVEC expression and chromatin accessibility were used. In other words, could this tool be used with solely publicly available data to improve variant interpretation within a population for which only population genetics/ selection data existed.
- 6.) Fig. 1E and 1F should show error bars.

Reviewer #2 (Remarks to the Author):

The manuscript by Xin et al. entitled "Chromatin accessibility landscape and regulatory network of high-altitude hypoxia adaptation" describes an ambitious experimental and bioinformatics effort to evaluate mechanisms of Tibetan high-altitude environmental adaptations that affect a wide variety of physiologic traits. Previous genome-wide analyses have determined that multiple genetic variants with strong selection signals lie within non-coding regions that include the locus encoding HIF-2 α (EPAS1), and other critical factors promoting these adaptations. The authors aim to generate a map of regulatory elements that impact gene expression, even though the cell type of action may be unknown, as well as relevant pathways, target genes, causality, and underlying mechanisms. They develop a variance interpretation model by Paired Expression and Chromatin Accessibility data (vPECA) to identify active regulatory elements (AREs) and active selected regulatory elements (ASREs), impacting associated gene regulatory networks. They demonstrate that genome-wide hypoxia regulatory networks operate via AREs and ASREs, and use these as a

resource to characterize the effect of genetic variants on high-altitude adaptation. Overall, this highly ambitious project is carefully conducted, and the conclusions are sound. I recommend acceptance upon addressing the following relatively minor concerns:

1. Most importantly, while the HUVEC cell types have been carefully chosen from individuals based on EPAS1 and EGLN1 genotypes, the culture conditions used have one important limitation. The authors appropriately chose to compare their assays in 21% O₂ versus 1% O₂ to identify hypoxic regulatory networks that could be relevant to extreme oxygen deprivation. This is a necessary but unavoidable limitation of their experimental set-up. Clearly, Tibetans do not live at 1% O₂, and instead exist at two-thirds 21% O₂ experienced by the Han Chinese. While I understand that this is needed to carry out the conducted studies, it is still a limitation and should be acknowledged as such in both the Results and Discussion sections.
2. Data provided on luciferase assays conducted in HEK293 and HeLa cells, while informative, do not really query the appropriate cell type for e.g. erythropoietin expression. These cells reside in the interstitium of the kidney and the assays may not reflect the physiologically relevant cell type in vivo. Again, there is nothing wrong with the experiments presented. It is a limitation that simply should be acknowledged, especially in the Discussion section.
3. Finally, while the results are very compelling, the Discussion should include speculation about future mouse models that would be necessary to functionally determine the in vitro findings described here.

Minor concerns:

1. Overall, the manuscript harbors numerous grammatical errors that should be carefully edited.
2. The authors cite reference no. 23 to describe oxygen signaling at consensus HREs. This work dates back to 2005, and has now been improved upon by the work of David Mole and Peter Ratcliffe in more recent papers. This should be revisited and updated accordingly.

Reviewer #3 (Remarks to the Author):

This work provided a rich data resource for open chromatin, gene expression and chromatin conformation profiles of Tibetan genome with hypoxia adaptation. With these datasets, they developed a novel variant interpretation method that prioritizes causal SNPs, and regulatory elements that drives essential genes responsible for hypoxia adaptation in the Tibetan population. In total, they generated 50 ATAC-seq and 50 RNA-seq for five Tibetan and five Han Chinese across five time points, 10 whole genome sequencing for the samples, and four Hi-C for selected four samples. The manuscript is relatively easy to follow, despite some grammar mistakes.

My major comments are listed below:

1. Data availability: the authors needs to provide processed data for the reviewer. Note: I am not asking for the raw reads, as I understand they might be sensitive data. I am asking the authors make the bigwig and Hi-C matrix available to the reviewer so that I can check the quality of their data.
 2. In Fig. 1B, the clustering analysis of gene expression shows that 0h, 6h, and 1d are separated better than the clustering based on ATAC-seq data. But the authors also suggested in previous sections that chromatin accessibility response earlier to hypoxia than gene expression.
 3. The authors identified 14 dynamically expressed TFs through motif search, however, the criteria for "dynamic" expression is not explained. This information is critical.
 4. How the EPAS1 RNAi experiments were done and how the RNAi result validates 621 EPAS1 target gene in the network were not explained.
- Please have the manuscript proof-read by professional writers. There are many grammar mistakes. One example is listed here: in L687, "don't consistent with."

Reviewer #4 (Remarks to the Author):

Comments on

Chromatin accessibility landscape and regulatory network of high-altitude hypoxia adaptation

Jingxue Xin et al Bing Su

This is really a timely and outstanding study. It should be publishable with only minor revisions. I have two questions.

1) Speaking of gene regulatory network, one usually refers to an interaction matrix with pairwise interactions represented by the off-diagonal elements and each gene's self-regulation is a diagonal element. As can be seen in Chen et al. (2019), the diagonal elements may play a larger role than the off-diagonal elements in stabilizing the gene regulatory network. I wonder what the authors may say about the diagonal elements in their system since they only mentioned pairwise interactions.

Ref: Yuxin Chen et al. 2019 Gene regulatory network stabilized by pervasive weak repressions: microRNA functions revealed by the May–Wigner theory. National Science Review (NSR) 0: 1–13, 2019 doi: 10.1093/nsr/nwz076 Advance access publication 0 2019

2) The main corresponding author has recently published a paper (also in NSR) which suggests that another gene in the same region, TMEN247, may be no less important than EPAS1. While the results on EPAS1 may not be all that surprising, TMEN 247 may need extra support. Any comments?

In short, I look forward to seeing this paper in print.

Point-to-point responses to **Reviewers' comments:**

Reviewer #1 (Remarks to the Author):

This manuscript presents a valiant effort to provide a biological interpretation for the signals of selection observed in Tibetan populations, particularly as they apply to the EPAS1 locus. To that end, they gather new data on transcript levels and chromatin dynamics in Tibetan and Han HUVECs exposed to hypoxia for increasing time lengths. Some of this data is then used in conjunction with an extension (vPECA) of a previously developed method to interpret the likely targets of selected regulatory elements in the Tibetan genome. This article presents a wealth of new data and data analysis that the authors rightly point out is key to better understanding and interpreting non-coding variation in human adaptation. However, this manuscript seems to fall apart in its attempt to apply their analytical approach to the EPAS1 locus. Several major concerns exist with regard to their experimental set up, the application of the method and the functional validation of regulatory elements. This article may hold promise as a resource or methods development paper, but it should not be suitable for publication in Nature Communications as written.

Author's Response: We are happy the reviewer appreciated our major contribution in statistical method vPECA and valuable data resources for better understanding and interpreting non-coding variation in human adaptation. In light of the reviewer's suggestion, we presented more genome-wide examples in addition to EPAS1 locus, carefully performed additional control experiments, and added functional validation in hypoxia conditions in our revision. We hope these revisions will address the major concerns and highlight our novel strategy to use hypoxia context-specific gene regulation to interpret variant besides method development and data resource.

Major concerns:

1.) The experimental set up for the RNAseq and ATACseq data is based on culturing cells in hypoxic conditions for increasing lengths of time including the 0h time point. Transcriptional or chromatin accessibility responses are then measured relative to different time points. This is not the standard set up for hypoxia response experiments where cells are cultured in parallel in normoxia and hypoxia for the same lengths of time. The experimental design used in this manuscript cannot distinguish between responses to hypoxia and responses to being in culture for different periods of time, which greatly complicates the interpretation of results.

Author's Response: We thank the reviewer's insightful comment to improve the experimental setup. We agree with the reviewer that our experimental design cannot distinguish between responses to hypoxia and being cultured in different periods of time. We performed additional experiments that cells are cultured in normoxia for the same lengths of time. We generated the RNA-seq data and analyzed it together with our previous RNA-seq data. We found that the responses to being in culture for different periods of time are not significant compared to the responses to hypoxia. Our main discovery and conclusions stand firmly.

In the new experiment, we collected 2 replicates of HUVEC from one Han Chinese and cultured them in normoxia (21% oxygen) for five days. Other cell culture conditions are controlled the same as previous experiments. RNA-seq was performed on 0h, 6h, 1d, 3d, and 5d for both replicates. Then

the raw data was processed following the same pipelines in the previous analysis. The total reads numbers and the alignment statistics of RNA-seq data are shown in Fig. R1A, indicating that the new data are in high quality. We visualized the high dimensional data in the two-dimensional plane by PCA. Two replicates tend to be closer and the main variance in PC1 is when cells being in culture for different periods of time in normoxia (Fig. R1B). This is consistent with the reviewer’s point that gene expression response to being in culture for different periods of time.

Figure R1. Alignment report and PCA of gene expression data when cells being cultured in normoxia for five days. (A) Raw reads number and alignment rates of all 10 samples. The samples are named with K8-replicate number-time. (B) PCA of FPKM expression data of cells cultured in normoxia for 5 time points. Two replicates are similar and cells are separated by different time periods.

Next, we compared the responses to being in culture for different periods of time to the responses to hypoxia. We pooled the data together, carefully removed the batch effect, and checked the variations of gene expression data between cells being cultured under normoxia and hypoxia for both the previous and new experiments. We used PCA to visualize the two datasets after batch effect adjustment (Fig. R2). The new samples (solid circles) are close to the previous 0h samples (red samples). The variance between normoxia samples in the new data (solid circles) is relatively smaller than the hypoxia ones (hollow shapes), especially in 3d and 5d. In the PCA plot, the largest variance revealed by PC1 is the response to hypoxia, i.e. , the difference between normoxia to hypoxia 1d and 3-5d under hypoxia (PC1 75%). This is the same as the results in Figure 1C. PC2 shows the difference between samples cultured in normoxia and hypoxia along time and is a mixture of response to culture time and hypoxia response. There are two key observations from the plot. 1) the responses to being in culture at normoxia for different periods of time are not significant compared to the responses to hypoxia. Although cells are different between 0h and 5d when cultured under normoxia, the variance is much smaller than 3d-5d under hypoxia (PC1 75% vs PC2 11%). Thus, the effect that cell responses to hypoxia is much larger than responses to being in culture at normoxia for different periods of time. 2) the response to culture along time is quite independent, dramatically different, and not related to the response to hypoxia in PC2. They have opposite directions that originated from the very similar 0h samples. This indicates there are two different gene groups underlying response to hypoxia and response to culture time.

Figure R2. PCA of gene expression data of previous and new experiments.

To further dissect the general patterns revealed by PCA, we compared the differentially expressed genes (fold change of the average value of two groups larger than 2) between different time periods under normoxia (new data, response to culture time) and hypoxia (previous data, response to hypoxia). The numbers of DEGs between adjacent time points in hypoxia response, especially 1d vs 3d and 3d vs 5d, are significantly larger than in normoxia by more than 30 folds (the number of DEGs are 1,000 vs 30 in Fig. R3A). In total, we identified 560 DEGs between 0h and 5d in normoxia with fold change > 2. The functional terms are significantly enriched in cell cycle and DNA replication (Fig. R3B, p -value < 10^{-40}), which are not related to hypoxia response.

Figure R3. Differentially expressed genes between hypoxia (previous data, response to hypoxia) and normoxia (new data, response to culture time). (A) The number of DEGs comparison between adjacent time points in two experiments. The red line indicates normoxia, while the blue line represents hypoxia. (B) Functional enrichment of 560 DEGs between 0h and 5d for response to cell culture along time.

To confirm the minor culture effect and DEG numbers observed in our new data, we also checked the public data in the GEO database for the impact of cell culture for different periods of time. We collected two independent datasets for HUVEC cultured in normoxia for 1h-24h and 0min-440min. We found cell culture along time did not have a dramatic effect on gene expression and cell culture effect is minor compared to the hypoxia effect. This is consistent with our experimental data and support that cell culture along time will not complicate our major findings. The detailed results are summarized as follows.

1. We collected the RNA-seq data of HUVEC in the GEO database with accession number GSE92506 (Huang, et al., 2017). This data contains static control (ST) samples for four time periods, 1, 4, 12, and 24 hours. Each time point contains 2 replicates. After downloading the raw reads count matrix, we calculated the expression FPKM value by the following formula.

$$FPKM = \frac{CDS \text{ read count} \times 10^9}{CDS \text{ length} \times \text{total mapped read count}}$$

where CDS is the coding sequence. We filtered out genes with FPKM 0. And we calculated the fold change between 1h and 24h in normoxia for each gene. Then we chose the cut off 2 for each replicate as differentially expressed genes. 45 genes consistently change in both replicates, which are identified as genes affected by 24h cell culture. These genes are not related to hypoxia response (GO:0001666).

2. We also collected the microarray data of HUVEC in the GEO database with accession number GSE21989 contributed by Di Camillo et al. HUVEC in the control group were cultured in 0, 40, 100, 200, 340, and 440min. After downloading the series matrix, we calculated the expression level of each coding gene as the maximum value of all probes. Then the data was normalized into log2 transformation. Similarly, we identified 103 DEGs between 0min and 44min by fold change larger than 1.2. These genes are related to endothelial cell proliferation. No gene is involved in response to hypoxia. We also tried fold change 2, and no gene was identified.

In addition, we searched the literature for the possible culture effect since HUVEC is a commonly used stable endothelial cell line in ENCODE and many other studies. A regenerative medicine study investigates “the ability of HUVECs to be expanded in culture to provide sufficient cells for graft seeding. The levels of gene expression of key genes are then examined to ensure that these cells retain

the EC phenotype. This study demonstrates that HUVECs may be cultured for up to 12 passages without alteration in phenotype” (Punshon, et al., 2011).

Taken together, several lines of evidence support that cell culture along time in normoxia will not affect our major findings. We can confidently draw the conclusion that the responses to being in culture for different periods of time are not significant, negligible, and independent of the responses to hypoxia. Our main discovery and conclusions stand firmly. We added the new data and results in our revision.

2.) With regard to the Hi-C data, it is well established that Hi-C without promoter capture identifies TAD boundaries, but does not have the resolution to identify actual enhancers. While the Hi-C data generated by the authors is valuable in its own right and may be useful in the PECA modeling, it cannot be used as a validation of specific enhancers, as presented – for example – in Fig. 3A. Incidentally, this figure is misleading because they do not represent other genes, besides EPAS1, that are also present within the same TAD (eg PRKCE). Also, it is unclear why in some analyses they use Hi-C data from previous publications and in others they use their own data.

Author’s Response: We agree with the reviewer that Hi-C without promoter capture identifies TAD boundaries but does not have the resolution to identify actual enhancers. In our study, the candidate enhancers in HUVEC are defined by the combination of H3K27ac plus H3K4me1, which is a commonly adopted strategy in many studies. The Hi-C data were used in our vPECA model to identify the TAD boundaries and to impose physical boundaries of enhancer-promoter (E-P) interactions. Fig. 3A clearly illustrates our strategy to define the regulatory boundary by HiC data and identify regulatory elements by histone modification data. We found that HiC information helps us greatly reduce the number of candidate E-P interactions. For example, there are 80 candidate enhancers within the 1M up- and down-stream region around EPAS1. After adding the Hi-C TAD boundary as shown in Fig. 3A, the number of candidate enhancers reduced to 23 in the TAD. This could reduce the number of parameters in vPECA model by ~4 folds.

We also agree with the reviewer’s point that Hi-C data does not have the resolution to identify actual enhancer-promoter (E-P) interactions. For example, in Fig. 3A, the Hi-C TAD around EPAS1 contains two genes including PRKCE. The Hi-C data cannot help us to identify real target genes and specific E-P interactions. We need paired ATAC-seq and RNA-seq to assess the correlation between enhancer activity and target gene expression and to reveal the true enhancer-target interactions.

Nevertheless, Hi-C data are still useful to provide evidence for long-range interactions predicted by vPECA. Since the resolution of the Hi-C data is 5kb, we used loops longer than 5k to identify E-P interactions. The enhancers are defined by the ATAC-seq peaks, and the regulation relations are detected from the Hi-C loops. We listed 3 examples, i.e. SOX17, SNAI2, and IL6, showing the E-P interactions detected by the Hi-C loops in Fig. R4. The E-P interactions are clearly revealed from the Hi-C loop data. Based on those observations, we think the loop data can partially provide physical interaction evidence to identify actual enhancer target relationships, but insufficient to detect them completely due to the resolution limit. We expect future data from Trac-looping, HiChIP, CHIA-PET technologies will provide E-P interactions with higher resolution.

Figure R4. E-P interactions detected by Hi-C loops.

In addition, we used the published Hi-C data since Rao et al. generated in situ Hi-C map for HUVEC at 5kb resolution, which shows high reproducibility with dilution Hi-C and contains many confident contacts. Later we generated our own HiC data and we found that they are quite conserved at topologically associating domains (TAD) level between different conditions (two time points and two populations) in our Hi-C data in HUVEC (Fig. R5). This is consistent with the observation in the 3D genome field that the TADs are conserved across a variety of cell types in the human genome (Pombo and Dillon, 2015). Therefore, the regulatory boundary information used in our model is conservative and may not change much whether we use public or our own Hi-C data. Moreover, one advantage of the strategy to use the public Hi-C TAD information is that vPECA could be applied to a more general context. After all, matched ATAC-seq and RNA-seq data can be easily accessible rather than the additional matched Hi-C data.

Figure R5. The number of TADs in our Hi-C data reveals the conservation of TAD. A (Adaptive) indicates Tibetan and W (Wildtype) is Han Chinese. 0 means 0h, and 3 means three days.

We also thank the reviewer's pointing the issue in Fig. 3A. We updated the figure by adding the PRKCE gene to this region both in the main text and listed below (Fig. R6). Interestingly, two elements (E7 and E8) are located in the PRKCE intron, but they are predicted to regulate EPAS1 (not PRKCE) by vPECA model and supported by the Hi-C read count under 3d hypoxia (Fig. 3D). These enhancers are within the same Hi-C loops with EPAS1's promoter but not PRKCE's promoter. Actually, the regulation of PRKCE is within the left loop linking PRKCE promoter and enhancers (Fig. R7). This indicates the advantage of using Hi-C loops as regulatory boundaries. Thus, in Fig. 3A we only show the regulation of EPAS1 within the two loops and put the regulation landscape of PRKCE and EPAS1 in the supplementary figure as fig. S6A.

Figure R6. Updated figure of Fig. 3A by adding PRKCE.

Figure R7. Regulation landscape of PRKCE and EPAS1.

Taken together, we agree with the reviewer’s comments on Hi-C. We updated and demonstrated our methods and results more clearly and with more details. We also noticed that advanced techniques, such as HiChIP, ChIA-PET, and Trac-looping, can provide more detailed regulation information between enhancers and target genes. We may generate these types of data to study hypoxia adaptation response in the future.

3.) No information is provided with regard to subjects that donated the umbilical blood cords other than the hospital and their self-reported ethnicity. Were they all born and raised at high altitude? Did the entire pregnancy take place at high altitude? Did they visit low altitude locations during the pregnancy or during the past 6 months? Transcript levels and chromatin conformation are molecular phenotypes that can be modified by environmental exposure especially in the context of hypoxic conditions. Was the ancestry of the donors verified by genetic analyses?

Author’s Response: In fact, we implemented stringent criteria, deep surveys, as well as genetic analyses in the sampling procedure to make sure that the ancestry of the donors are Tibetans. We added that information in our revision to the main text and we thank the reviewer to improve our manuscript.

Firstly, the ancestry of the donors was determined by the self-report and verified by the genetic analyses using genotyping and whole genome sequencing data. Only the Tibetans, whose lineal relatives are also Tibetan ethnicity within three generations, were included in this study. The same strategy was also implemented for Han donors.

For genetic verification, we chose the adaptive (Tibetan) and wildtype (Han Chinese) populations according to the EPAS1/EGLN1 genotypes using Tibetan-specific common SNPs (Table R1). Given the DAF (derived allele frequency) of these SNPs in the Han population is extremely low (near 0), we can determine ancestry exactly using their genotypes. Additionally, we used known ancestry populations conducted principal component analysis, which also supports the ancestry identification from genotypes.

For pregnancy situation, we confirmed that the Tibetan donors were all born and raised at high altitude (Lhasa, elevation: 3680m), and their entire pregnancy procedure took place at high altitude and no low altitude visit during the period. Biologically related individuals were excluded from this study. Written informed consents for all participants were obtained.

Table R1. Tibetan- specific SNPs for ancestry characterization.

SNPs	Han	Tibetan	ancestral allele	derived allele	DAF-TBN	DAF-HAN
rs73926265	G G	A A	G	A	0.777	0.005
rs149594770	T T	A A	T	A	0.572	0.005
rs186996510	G G	C C	G	C	0.700	0.034
TED(3.4kb)	Insertion	Deletion	Insertion	Deletion	0.500	0.000

4.) While the authors focus on a single SNP (rs3768729) and region (E22-1) which appear to show evidence of reduced enhancer activity and chromatin openness in the Tibetan-enriched haplotype, they fail to discuss other regions on the same tightly linked haplotype which show the exact inverse patterns in their analysis. For example, the SNPs in E20 consistently show the inverse effect (the wildtype allele significantly

decreases enhancer activity and is associated with reduced chromatin openness). Within E22, the two regions/SNPs tested appear to go in opposite directions with regards to the adaptive alleles' association with chromatin openness. While these may not entirely invalidate the overall conclusions regarding multiple enhancers contributing to the downregulation of EPAS1 in Tibetans, they need to be addressed rather than buried in the supplement.

Author's Response: Thanks for the reviewer's crucial comments. We agree that except for E22-1, other regions on the same tightly linked Tibetan-enriched haplotype should be discussed in the main text rather than buried in the supplement.

As the reviewer suggested, E20 and E22 show the exact inverse patterns of chromatin accessibility between two populations. It may indicate the complexity of gene regulation. Usually, gene expression is regulated by multiple regulatory elements under certain temporal and spatial conditions. The regulatory elements, binding with various transcription factors, may activate or repress gene expression. Currently, to our limited knowledge of the regulation of EPAS1 expression, it is down-regulated in Tibetan than Han Chinese (Fig. 3B). There are 23 regulatory elements (REs) potentially regulating its expression. The overall effect of all enhancers is down-regulation in Tibetan, although there might be some REs with the opposite trend. In general, among all 23 REs, the openness scores of 7 (p-value<0.1) and 3 (p-value<0.05) elements of Tibetan are smaller than Han. Only the openness scores of E7 in Tibetan are slightly larger than Han with p=0.11 (fig. S6B, i.e., Fig. R8). Therefore, when pooling all the enhancers together, the overall regulation output is decreased gene expression.

Figure R8. Chromatin accessibility dynamics for the 23 candidate regulatory elements of EPAS1.

When considering the effect of multiple SNPs in a LD region, although some SNPs might have an inversed activity pattern such as E20 and E22, the integrative effect is down-regulation in Tibetan. If we aggregate all SNPs in Fig. 3E, the activity of reporter assay in Tibetan is lower than Han.

Another observation is that E20 and E22 are functional at different time, as shown in Fig. 3B. The activity of E20 increases at 6h, while E22 at 5d. Thus, the two enhancers may function in early and late-stage, respectively under hypoxia.

Taken together, EPAS1 is regulated by multiple enhancers, whose activity is different along time and affected by various SNPs in the tightly linked haplotype. Although the functions and effect sizes of

these SNPs and enhancers might be different, the overall output is down-regulation of EPAS1 in Tibetan than Han Chinese. We added the above explanation to the Result section in our revision.

5.) rs3768729 which the authors describe as having a striking effect on enhancer activity, and putatively EPAS1 expression in Tibetans based on their reporter gene assays is a common SNP worldwide. The putatively adaptive allele is ancestral and segregates at .48-.58 frequency globally. If this SNP had such a profound effect on enhancer activity and EPAS1 expression, it is surprising that it has not been detected in any GWAS or eQTL study given this high frequency. The authors should address this point if they wish to put this SNP forward as the most probable and/or important causal SNP under selection at the EPAS1 locus.

Author's Response: Yes, the Tibetan-enriched allele (putatively adaptive allele) is the ancestral allele "T" for rs3768729, and the T allele reaches 85% in Tibetans with lower frequencies in other world populations (Fig. R9). In particular, there is a deep divergence of rs3768729 between Tibetans and Han Chinese ($F_{ST}=0.52$), suggesting selection on the ancestral allele. The relatively high ancestral allele frequency in Europeans (0.65) might be caused by genetic drift rather than selection. Additionally, according to our previous study (Peng, et al., 2017), EPAS1 displays its function by the Tibetan-specific haplotype, not just a variant, and selection signal of rs3768729-T may result from a hitchhiking effect. Finally, the E22 enhancer where rs3768729 is located exhibited activity change in hypoxia (5d) while not in normoxia (0h) (Fig. 3C lower right), suggesting that rs3768729-T may only work in hypoxic condition. We also performed an additional reporter gene assay experiment after 36h incubation in 1% oxygen hypoxia environment. Under hypoxia, the activity differences for E22-1 (containing rs3768729) between the Han version and the Tibetan version became larger ($p = 0.0038$ in HEK293 and $p=3.51 \times 10^{-5}$ in HELA). Taken together, this could explain why rs3768729-T was not detected in the published eQTL data in spite of high frequencies in global populations.

Figure R9. Allele frequencies of rs3768729 in global populations.

Figure R10. Dual-luciferase enhancer reporter gene assay of E22-1 (rs3768729). All assays were performed in at least three replicates and the p-values were calculated by t-test.

We add those discussions and the new functional experimental results for SNP rs3768729 in our revision (Figure 3 in the main text and fig S7 in supplementary materials). We thank the reviewer’s insightful comment to improve the causal SNP interpretation.

6.) The choice of cell type, while interesting, is not well justified. The authors claim the endothelium is hypoxia-responsive and therefore a good choice of tissue in which to assess Tibetan chromatin structure. This statement seems to assume either that the findings within the endothelium are representative of other hypoxia-responsive tissue types (which is unlikely, given that regulatory networks will vary between tissues) or that endothelium is the most relevant tissue type in which to examine hypoxia adaptation. If the latter, the authors need to do a better job of explaining this choice.

Author’s Response: We agree with the reviewer that we need to better explain why endothelium is the most relevant cell type for hypoxia study. In the revised version, we added a subsection to give more explanations in the Methods (“Choosing relevant cell type to hypoxia response” section). The followings are the presented evidence for choosing HUVEC.

1. HUVEC can be easily accessed compared with other tissues with SNP enrichment signals in Figure R11, such as ovary, pancreas, and brain etc. First, we agree with the reviewer’s opinion that findings within the endothelium aren’t representative of other hypoxia-responsive tissue types because of tissue specificity. We know that hypoxia adaptation is a complex physical trait that involves multiple tissues in the respiratory and circulatory systems. Other tissues are also important when interpreting selection signals. But under the current condition, we were not able to get access to other relevant tissues, such as lung and heart in human. Second, although gene regulatory networks are different between tissues, hypoxic inducible pathways are conserved to some extent. The HIF family genes, such as HIF1A and EPAS1, have similar expression patterns responsive to hypoxia. Thus, we could only try our best to interpret part of the selection variants falling into the open regions in HUVEC and leave the other relevant tissues for future studies.
2. HUVEC is a classic model to study oxidative stress and hypoxia, due to its oxygen-sensitive and easy to obtain. We found many references using it to study cellular responses to hypoxia (Cao, et al., 2019; Michiels, et al., 2000; Namiki, et al., 1995; Therade-Matharan, et al., 2004). Nakato et al. recently cataloged gene expression and active histone marks in nine types of human ECs (generating 148 genome-wide datasets) and carried out a comprehensive analysis with chromatin

interaction data. They pointed out that endothelial cells (ECs), which make up the innermost blood vessel lining of the body, express diverse phenotypes that affect their morphology, physiological function and gene expression patterns in response to the extracellular environment, including the oxygen concentration, blood pressure and physiological stress (Nakato, et al., 2019).

3. HUVEC is listed as the Tier 2 ENCODE Project common cell types (<https://www.genome.gov/encode-project-common-cell-types>). Cell types were selected largely for practical reasons, including their wide availability, the ability to grow them easily, and their capacity to produce sufficient numbers of cells for use in all technologies being used by ENCODE investigators. Importantly, HUVEC has a normal karyotype and are readily expandable to 10⁸-10⁹ cells. ENCODE and NCBI GEO database collected rich public data for HUVEC, i.e., RNA-seq, DNase-seq, and ChIP-seq experiments, which provide rich independent validation samples.
4. SNPs under selection in Tibetans are enriched in the open chromatin regions in HUVEC (Fig 1F & S3C, i.e., Fig. R11). The fold enrichment is 1.5 for normoxia and 1.6 for hypoxia. This demonstrates that HUVEC is sensitive to hypoxia. We also observed strong selection SNPs located in the open regions of HUVEC (fig. S8E) around EPAS1, the major effect gene in adaption to hypoxia in Tibetans. These observations together provide the feasibility to utilize the regulatory regions in HUVEC to interpret functional variants in Tibetans.

Figure R11. SNPs under selection are highly enriched in HUVEC and a variety of cell types/tissues from ROADMAP. The FC score is calculated across 40 DNase-seq samples from ROADMAP (Supplementary methods). More stringent thresholds give higher fold enrichment scores.

7.) As far as I can tell, the reporter gene assays were performed in normoxic conditions where both HIF1A and HIF2A are tagged for ubiquitination and degraded. Therefore, these are not the appropriate conditions for validating the enhancers and SNP effects and the results highly questionable.

Author's Response: To address this question, we performed additional gene reporter assay experiments under hypoxia in both HEK293 and HELA, and the results are shown in Fig. R12 and also updated in Fig. 3E. The cells were incubated for 36h under hypoxia after transfection, and the other experimental conditions were the same as the previous assays under normoxia.

Figure R12. Dual-luciferase reporter gene assay of five upstream EPAS1 variants at normoxia and hypoxia culture conditions of 37 °C in two cell lines HEK293 (left) and HELA (right). All assays were performed in at least three replicates and the p-values were calculated by t-test.

The 4 REs (E12, E20, E21, and E22) are in general more active under hypoxia except for E21-2, suggesting their involvement in response to hypoxia by regulating EPAS1 expression. Notably, under hypoxia, the enhancer activity differences between the Han version and the Tibetan version became larger for E20 ($p = 0.05$ in HEK293), E22-1 ($p = 0.0038$ in HEK293 and $p = 3.51 \times 10^{-5}$ in HELA), and E21-2 ($p = 0.01$ in HELA). This strongly supports the proposed functional roles of these adaptive SNPs (rs370299814, rs368706892, rs3768729 and rs6756667).

Taken together, our reporter assay experiments in both normoxic and hypoxic conditions are effective in detecting functional REs that contribute to the adaptive expression regulation of EPAS1 in Tibetans. The above results were added to our revision.

8.) The assess the significance of the composite multiple signals the authors combine p-values using Fisher's approach (lines 742-745), but this approach assumes that the p values from independent tests of the same hypothesis, which is clearly not the case here because SNPs are in LD thus not independent.

Author's Response: We agree that Fisher's approach assumes that the p-values from independent tests of the same hypothesis. Here we used Fisher's method to integrate the independent selection signals from different population genetic tests for a single SNP. This task will not consider many SNPs and should not be affected by the LD structure. The fact that SNPs in LD are not independent is considered in the first equation of vPECA estimating the selection status of each regulatory element.

Four selection scores, i.e. Fst, iHS, XP-EHH, and PBS, were composited together by Fisher's method. An overall p-value for each SNP is calculated by chi-squared distribution. This method is a simplified version of CMS score by a composite of multiple selection signals (Grossman, et al., 2010). The original CMS method treats multiple tests for selection for each SNP as independent. Thus, the CMS score is the posterior probability that the SNP is selected, which can be decomposed into the product of the probability of each individual test. This is why we used Fisher's method to combine different selection scores by assuming them as independent. Therefore, for every single SNP, the screening method calculates one score.

We consider the LD effect in the first equation of vPECA estimating the selection status of each regulatory element, i.e.,

$$\log \frac{P(S_k = 1|X, Y)}{1 - P(S_k = 1|X, Y)} = \mu_0 + \sum_{q=1}^Q \mu_q \sum_{p \in J_k} \frac{1}{|J_k|} w_p X_{p,q} + \mu_{Q+1} \sum_{p \in J_k} \frac{1}{|J_k|} w_p Y_p,$$

where w_p indicates the weight of the p-th SNP, which is the reciprocal of the LD score of SNP p. This term can down-weight the influence of SNPs within a large LD, which reduces the redundancy of correlated SNPs. The idea was referred from a widely used selection detection method XP-CLR (Chen, et al., 2010; Grossman, et al., 2010).

9.) The project accession PRJNA544821 is not found in Bioproject.

Author's Response: In the revision, the data generated in this study, including RNA-seq, ATAC-seq, and Hi-C data were deposited at <http://www.ncbi.nlm.nih.gov/geo/> (accession number: GSE145774) and Genome Sequence Achieve (project number: CRA002025; <https://bigd.big.ac.cn/gsa/browse/CRA002025>). A reviewer link has been generated as follows: <https://www.ncbi.nlm.nih.gov/geo/query/acc.cgi?token=cnylueyszfqxnm&acc=GSE145774>. The source codes for vPECA are available at <https://github.com/AMSSwanglab/vPECA>.

Minor Comments:

1.) The putatively selected regulatory regions defined by vPECA are typically much larger (>1kb) than the regions looked at for differential openness (~200-550bp) which are again larger than the very short segments examined in the luciferase assays (100bp).

Author's Response: We thank the reviewer's question. The putatively selected regulatory regions defined by vPECA (based on 3 histone modification marks by ChIP-seq data) is much larger than the short segments we used in luciferase assays, which are about 200-550bp long (fig. S8A). The choice of short segments is based on narrow ATAC-Seq peaks located in the broad regulatory regions (usually several thousand bps) where variants under selection are located. This design helps to study 1-2 variants separately in short segments. Also, the technical limitation of luciferase assay usually requires short segments no longer than several hundred base pairs.

2.) The cell types chosen for the luciferase assays re not endothelial and enhancer activity in these contexts may not be representative of the endothelial regulatory network.

Author's Response: We acknowledged HEK293 and HeLa cells may not reflect the physiologically relevant cell type in vivo and we discussed the potential limitation in the Discussion sections in our revision. Our choice of cell types is based on the following thoughts.

Excerpt from Manuscript: We used HEK293 and HELA to do a dual-fluorescent reporting system just to verify the elements are functional as enhancers. The transfection of the dual-fluorescent reporting system is usually transient, so the dual-fluorescent reporting system often uses cell lines of high transfection efficiency, easy culture and rapid growth such as HEK293 and HELA. The transfection efficiency of the primary culture HUVEC is less than 10%, and our current culture system only allows the generation of the HUVEC for 92h. Therefore, the signals could not be detected by the dual-fluorescence reporting system. In order to reflect the physiologically relevant cell type in

vivo, the direct evidence is knock-out and knock-in by CRISPR-Cas9 system. We plan to establish the immortal HUVEC cell line and conduct gene editing in the future.

3.) In lines 360-361 the authors describe RORA as having “the second strongest signal for high-altitude adaptation” in Ethiopian populations. However, the articles they cite for this claim explicitly state that while a SNP in RORA is the second strongest signal of association with Hb concentration in the Amhara, it has no signal of selection in their data. Therefore, the statement above is false or at least misleading.

Author’s Response: We thank the reviewer for pointing out this inaccurate citation. We revised this sentence to “We note that a SNP in RORA is the second strongest signal of association with Hb concentration in the Amhara in Ethiopia” in the manuscript.

4.) The paper as written is littered with typos that would need to be fixed in a final edit.

Author’s Response: We thank the reviewer’s comment. We carefully checked and corrected the typos in the main manuscript.

5.) While the Authors present a compelling data analysis pipeline, they appear to apply this model specifically to their Tibetan expression and chromatin accessibility data, while using publicly available data for the other aspects of the model. For the purposes of method development, it would be interesting to determine how similar the results would be if, for instance, ENCODE data on HUVEC expression and chromatin accessibility were used. In other words, could this tool be used with solely publicly available data to improve variant interpretation within a population for which only population genetics/ selection data existed.

Author’s Response: We thank the reviewer’s insightful question and we appreciate the reviewer’s interest in the general usage of our method. The short answer is yes. In our previous publication (Duren et al, PNAS, 2017, <https://www.pnas.org/content/114/25/E4914>), we demonstrated that we used solely publicly available ENCODE data to improve variant interpretation for which only population genetics/ selection data existed. There the regulatory model inferred from accessibility and expression data in diverse contexts from mouse ENCODE database provided interpretation for QTLs in MGI database. The software and results can be accessed at <https://github.com/SUwonglab/PECA>.

However, we recommend integrating context relevant and specific omics data with public data as demonstrated in our manuscript. This will greatly improve the confidence of genetic interpretation. As we showed in this manuscript, hypoxia condition is critical to reveal the selected SNPs’ function and it’s necessary to generate the omics data in hypoxia condition.

6.) Fig. 1E and 1F should show error bars.

Author’s Response: We thank the reviewer’s comment to improve the figures. We updated Fig. 1F with error bars, which indicate the standard error of fold change between replicates.

Fig. 1E shows the number of differentially expressed genes (DEG) between every two adjacent time points. We chose $FDR < 0.05$ as the cutoff. The numbers of DEGs are constant but not random variables. Therefore, we can’t plot error bars in this figure.

Reviewer #2 (Remarks to the Author):

The manuscript by Xin et al. entitled “Chromatin accessibility landscape and regulatory network of high-altitude hypoxia adaptation” describes an ambitious experimental and bioinformatics effort to evaluate mechanisms of Tibetan high-altitude environmental adaptations that affect a wide variety of physiologic traits. Previous genome-wide analyses have determined that multiple genetic variants with strong selection signals lie within non-coding regions that include the locus encoding HIF-2 α (EPAS1), and other critical factors promoting these adaptations. The authors aim to generate a map of regulatory elements that impact gene expression, even though the cell type of action may be unknown, as well as relevant pathways, target genes, causality, and underlying mechanisms. They develop a variance interpretation model by Paired Expression and Chromatin Accessibility data (vPECA) to identify active regulatory elements (AREs) and active selected regulatory elements (ASREs), impacting associated gene regulatory networks. They demonstrate that genome-wide hypoxia regulatory networks operate via AREs and ASREs, and use these as a resource to characterize the effect of genetic variants on high-altitude adaptation. Overall, this highly ambitious project is carefully conducted, and the conclusions are sound. I recommend acceptance upon addressing the following relatively minor concerns:

Author’s Response: We are happy the reviewer appreciated our major contribution for a variant interpretation model vPECA and data resources of genome-wide hypoxia regulatory networks to characterize the effect of genetic variants on high-altitude adaptation.

1. Most importantly, while the HUVEC cell types have been carefully chosen from individuals based on EPAS1 and EGLN1 genotypes, the culture conditions used have one important limitation. The authors appropriately chose to compare their assays in 21% O₂ versus 1% O₂ to identify hypoxic regulatory networks that could be relevant to extreme oxygen deprivation. This is a necessary but unavoidable limitation of their experimental set-up. Clearly, Tibetans do not live at 1% O₂, and instead exist at two-thirds 21% O₂ experienced by the Han Chinese. While I understand that this is needed to carry out the conducted studies, it is still a limitation and should be acknowledged as such in both the Results and Discussion sections.

Author’s Response: We thank the reviewer to discuss the possible limitation on the culture condition. This motivates us to discuss the potential limitation in the Methods section in our revision. We acknowledged that the choice of 1% O₂ is empirical and mainly based on the literature (Antonova, et al., 2014; Casanello, et al., 2005; Collard, et al., 1999; Graham, et al., 1998; Kumar, et al., 2011; Losenkova, et al., 2018; Monteiro, et al., 2019; Schmedtje, et al., 1997; Shimpuku, et al., 2000; Suzuki, et al., 1999; Voellenkle, et al., 2016; Wu, et al., 2003; Zhang, et al., 2016). Our choice of 1% O₂ is roughly estimated based on the following understandings.

Excerpt from Manuscript: Oxygen partial pressures of cells and bodies are different. Oxygen is inhaled by the body to reach the mitochondria of the cells, and the partial pressure of oxygen is a progressive step down. At sea level (normoxia), oxygen partial pressure is 159.22 mm hg (20.95%), but the partial arterial oxygen pressure (PaO₂) is only 100 mmHg (13%), while the venous oxygen partial pressure (PvO₂) is 40 mmHg (5.2%), and the mitochondrial oxygen partial pressure is 4 to 20 mmHg (0.52% to 2.6%).

2. Data provided on luciferase assays conducted in HEK293 and HeLa cells, while informative, do not really query the appropriate cell type for e.g. erythropoietin expression. These cells reside in the interstitium of the kidney and the assays may not reflect the physiologically relevant cell type in vivo. Again, there is

nothing wrong with the experiments presented. It is a limitation that simply should be acknowledged, especially in the Discussion section.

Author's Response: We acknowledged that HEK293 and HeLa cells may not reflect the physiologically relevant cell type *in vivo* and we discussed the potential limitation in the Methods section in our revision. Our choice of cell types is based on the following thoughts.

Excerpt from Manuscript: We used HEK293 and HELA to do a dual-fluorescent reporting system just to verify the elements are functional as enhancers. The transfection of the dual-fluorescent reporting system is usually transient, so the dual-fluorescent reporting system often uses cell lines of high transfection efficiency, easy culture and rapid growth such as HEK293 and HELA. The transfection efficiency of the primary culture HUVEC is less than 10%, and our current culture system only allows the generation of the HUVEC for 92h. Therefore, the signals could not be detected by the dual-fluorescence reporting system. In order to reflect the physiologically relevant cell type *in vivo*, the direct evidence is knocked-out and knocked-in by CRISPR-Cas9 system. We plan to establish the immortal HUVEC cell lines and conduct gene editing in the future.

3. Finally, while the results are very compelling, the Discussion should include speculation about future mouse models that would be necessary to functionally determine the *in vitro* findings described here.

Author's Response: We thank the reviewer to point out the future mouse models are necessary to functionally determine the *in vitro* findings we made. We added this as the future work in the Discussion sections in our revision.

Indeed, the enhancers (containing SNP or not) to be functional verified could be site-directed mutated or knocked out by CRISPR-Cas9 system to construct mouse models for studying the physiological phenotypes.

Minor concerns:

1. Overall, the manuscript harbors numerous grammatical errors that should be carefully edited.

Author's Response: We thank the reviewer's comment. We carefully checked and corrected the grammar mistakes in the manuscript.

2. The authors cite reference no. 23 to describe oxygen signaling at consensus HREs. This work dates back to 2005, and has now been improved upon by the work of David Mole and Peter Ratcliffe in more recent papers. This should be revisited and updated accordingly.

Author's Response: We thank the reviewer's suggestion for the improved work by David Mole and Peter Ratcliffe. We have updated two recent papers (Platt, et al., 2016; Smythies, et al., 2019) in the references.

Reviewer #3 (Remarks to the Author):

This work provided a rich data resource for open chromatin, gene expression and chromatin conformation profiles of Tibetan genome with hypoxia adaptation. With these datasets, they developed a novel variant interpretation method that prioritizes causal SNPs, and regulatory elements that drives essential genes responsible for hypoxia adaptation in the Tibetan population. In total, they generated 50 ATAC-seq and 50 RNA-seq for five Tibetan and five Han Chinese across five time points, 10 whole genome sequencing for the samples, and four Hi-C for selected four samples. The manuscript is relatively easy to follow, despite some grammar mistakes.

Author's Response: We appreciated the reviewer's positive comments for the rich multi-omics data resource we provided for Tibetan hypoxia adaptation, and the novel variant interpretation method proposed to prioritize causal SNPs using gene regulatory network.

My major comments are listed below:

Data availability: the authors needs to provide processed data for the reviewer. Note: I am not asking for the raw reads, as I understand they might be sensitive data. I am asking the authors make the bigwig and Hi-C matrix available to the reviewer so that I can check the quality of their data.

Author's Response: We totally agree that this friendly data sharing will increase vPECA's impact. In the revision, the data generated in this study, including RNA-seq, ATAC-seq, and Hi-C data were deposited at <http://www.ncbi.nlm.nih.gov/geo/> (accession number: GSE145774) and Genome Sequence Achieve (project number: CRA002025; <https://bigd.big.ac.cn/gsa/browse/CRA002025>).

A reviewer link has been generated as

follows: <https://www.ncbi.nlm.nih.gov/geo/query/acc.cgi?token=cnylueyszfqxnm&acc=GSE145774>.

The source codes for vPECA are available at <https://github.com/AMSSwanglab/vPECA>.

There are three essential components for the data organization including a metadata spreadsheet, the processed data files, and raw data files. The processed data include variant calling vcf format file, hg19 alignment bam and bigwig files for ATAC-seq and RNA-seq, bed format peak files of ATAC-seq, FPKM gene expression matrix, .hic files and loops for Hi-C data. The data have been released. The reviewers can easily get access to the data via ftp://download.big.ac.cn/gsa/CRA002025/processed_data/

2. In Fig. 1B, the clustering analysis of gene expression shows that 0h, 6h, and 1d are separated better than the clustering based on ATAC-seq data. But the authors also suggested in previous sections that chromatin accessibility response earlier to hypoxia than gene expression.

Author's Response: We agree that 0h, 6h, and 1d are separated in gene expression better than the clustering based on ATAC-seq data. So only the clustering result is not sufficient to claim the conclusion that chromatin accessibility response earlier to hypoxia than gene expression. Therefore, we added more evidence to support this finding.

First, we compared the number of differentially expressed genes (DEGs) and open regions between 0h and 6h. We identified 517 DEGs (out of 12,998 genes in total, and the proportion is 3.98%) and 8,551 (out of all 54,102 regions, the proportion is 15.81%) differential open regions (DORs) between normoxia 0h and hypoxia 6h with FDR<0.05. Thus between 0h and 6h, we identified a much larger proportion of DORs than DEGs (by 4 folds), which suggests after hypoxic induction for 6h, the chromatin accessibility of regulatory elements changes more dramatically than gene expression. This suggests chromatin accessibility response earlier to hypoxia than gene expression.

Second, the regulation of EPAS1 is one of the examples clearly indicating that changing RE's accessibility is earlier than gene expression. In Fig. R13 (i.e. Fig. 3B&C), the chromatin accessibility of E20 and E21 increases dramatically from 0h to 6h, and then EPAS1 expression begins to increase from 1d.

Figure R13. Gene expression and RE accessibility of EPAS1.

Third, there is further evidence from literature, TFs and specific chromatin remodelers often facilitate higher-order chromatin organization and regulate access to DNA (Klemm, et al., 2019). The remodeling process changes the open chromatin state of regulatory elements, which regulate the transcription of neighboring genes. Therefore, it is reasonable to observe chromatin accessibility dynamics before the change of gene expression.

3. The authors identified 14 dynamically expressed TFs through motif search, however, the criteria for “dynamic” expression is not explained. This information is critical.

Author's Response: We thank the reviewer's question. We supplemented the detailed approach in “Motif scan and identification of enriched dynamic transcription factors” in the Methods section.

Excerpt from Manuscript: Fig. 1D shows the 14 TFs selected by dynamic expression and motif enrichment. In the figure, the color of each dot represents the average gene expression value and the circle size indicates the average ME score among all samples at each time point. The gene expression FPKM values are divided into 5 levels, i.e. Level I: < 6, Level II: 6-12, Level III: 12-30, Level IV: 30-120, and Level V : > 120, which are labeled by 5 colors. Similarly, the ME scores are divided into 7 levels, i.e. Level I: < 2, Level II: 2-4, Level III: 4-10, Level IV: 10-50, Level V: 50-90, Level VI: 90-240, and Level VII : > 240, showing by the circle size. Then we filtered out motifs by two requirements, i.e. (1) Maximal FPKM ≥ 12 , and (2) Maximal ME Score ≥ 2 , which gives 43 TFs. Among these TFs, we also required the TFs should be dynamic, i.e. the expression FPKM and ME scores are not in the same Level (Level I-V for gene expression and Level I-VII for ME score) along 5 time points, thus obtaining 14 TFs in the short list.

4. How the EPAS1 RNAi experiments were done and how the RNAi result validates 621 EPAS1 target gene in the network were not explained.

Author's Response: We thank the reviewer's question. We have added a new subsection titled "Validation of 621 target genes of EPAS1 by siRNA knockdown experiments" in the Methods part.

Excerpt from Manuscript: In detail, the cell lines of HUVEC and C166 were transfected with EPAS1 siRNA and non-targeting control siRNA for 48 hours. The efficiencies of knocking down the EPAS1 expression were assessed by qPCR with 1.4% for HUVEC and 3.2% for C166 (Yoo, et al., 2015). The FPKM gene expression data of HUVEC and C166 was downloaded from the GEO database with accession number GSE62974. Then EPAS1's target genes were derived by comparing expression profiles of EPAS1 siRNAs with non-targeting ones. We calculated the fold change (FC) of average FPKM values between replicates and identified 795 genes in HUVEC and 1,382 in C166 with FC larger than 1.5. These genes were assumed to be true positive to validate 621 EPAS1 target genes derived from the regulatory network. Then we used the hypergeometric test to calculate the enrichment of vPECA predicted target genes of EPAS1 in the gold standard gene set. The hypergeometric cumulative distribution function is given by

$$F(x|K, M, N) = \sum_{i=0}^x \frac{\binom{K}{i} \binom{M-K}{N-i}}{\binom{M}{N}},$$

where x is the number of predicted target genes of EPAS1 in the gold-standard gene set. M denotes the total number of expressed genes in HUVEC. K indicates the number of gold standard genes expressed in HUVEC. And N is the number of predicted EPAS1 target genes. The hypergeometric test p-values are 1.22×10^{-4} in HUVEC and 7.8×10^{-11} in C166, which indicates vPECA predicted target genes of EPAS1 are significantly enriched in the differentially expressed genes in the knockdown experiment of HUVEC and C166.

Please have the manuscript proof-read by professional writers. There are many grammar mistakes. One example is listed here: in L687, "don't consistent with."

Author's Response: We thank the reviewer for pointing out the grammar mistakes. We have corrected the grammar errors in the manuscript.

Reviewer #4 (Remarks to the Author):

Comments on

Chromatin accessibility landscape and regulatory network of high-altitude hypoxia adaptation

Jingxue Xin et al Bing Su

This is really a timely and outstanding study. It should be publishable with only minor revisions. I have two questions.

1) Speaking of gene regulatory network, one usually refers to an interaction matrix with pairwise interactions represented by the off-diagonal elements and each gene's self-regulation is a diagonal element. As can be seen in Chen et al. (2019), the diagonal elements may play a larger role than the off-diagonal elements in stabilizing the gene regulatory network. I wonder what the authors may say about the diagonal elements in their system since they only mentioned pairwise interactions.

Ref: Yuxin Chen et al. 2019 Gene regulatory network stabilized by pervasive weak repressions: microRNA functions revealed by the May–Wigner theory. National Science Review (NSR)

0: 1–13, 2019 doi: 10.1093/nsr/nwz076 Advance access publication 0 2019

Author's Response: We agree that gene's self-regulation is important in mRNA degradation and stabilizing gene regulatory network. It has been nicely demonstrated that self-regulation may play a larger role than the off-diagonal elements in stabilizing the GRN in the published study (Chen, et al., 2019). However, our vPECA model has the limitation to infer the self-regulation relations due to the Bayesian inference structure. When modeling expression of a target gene (TG) given a set of candidate TFs, we used the conditional density, i.e. $P(TG|TFs)$. For self-regulation situations, TG is also a TF and there will be a self-loop. It violates the assumption that the Bayesian network should be based on a directed acyclic graph. Therefore, our method could not infer the self-regulation structure. It can be extended to dynamical Bayesian network to handle self-regulation, which we plan to try in future studies.

We cited the reference of Chen et al (2019), pointing out the importance of self-regulation, and discussed the potential limitation of vPECA in the Discussion section.

2) The main corresponding author has recently published a paper (also in NSR) which suggests that another gene in the same region, TMEM247, may be no less important than EPAS1. While the results on EPAS1 may not be all that surprising, TMEM247 may need extra support. Any comments?

In short, I look forward to seeing this paper in print.

Author's Response: TMEM247 is another important gene with strong selection signals in Tibetans. TMEM247-rs116983452 has recently been identified to be significantly correlated with reduced hemoglobin concentration, red blood cell count, and hematocrit (Deng, et al., 2019). However, this gene's expression is highly context-specific and is not expressed in HUVEC (expression value is 0 across all the 50 samples). This makes it hard to interpret the regulatory mechanisms under our current dataset.

We also checked the expression profile of TMEM247 in ENCODE, GTEx, and gene expression data of mammalian (human and macaque) organ development (Cardoso-Moreira, et al., 2019), which is shown in the following figures (Fig. R14-16). These expression profiles show TMEM247 mainly expresses in testis, implying its potential function in male reproduction, which is worthy to be studied in the future. We added those new data and discussions in our revision. We thank the reviewer's insights to improve our manuscript.

Courtesy of ENCODE

Fig. R14. The expression profile of TMEM27 in ENCODE shows that TMEM27 is expressed in testis, blood, and embryo but not in other tissues or cell lines.

Fig. R15. The expression profile of TMEM27 in multiple tissues in GTEx shows that TMEM27 is highly expressed in testis, but mostly absent in other 40 tissues.

Fig. R16. Expression profile of TMEM247 during development across 6 major tissues.

Reference

- Antonova, L.V., *et al.* [Proliferative and secretory activity of human umbilical vein endothelial cells cultured under varying degrees of hypoxia]. *Tsitologiya* 2014;56(1):67-76.
- Cao, H., *et al.* Hypoxia destroys the microstructure of microtubules and causes dysfunction of endothelial cells via the PI3K/Stathmin1 pathway. *Cell Biosci* 2019;9:20.
- Cardoso-Moreira, M., *et al.* Gene expression across mammalian organ development. *Nature* 2019;571(7766):505-509.
- Casanello, P., *et al.* Equilibrative nucleoside transporter 1 expression is downregulated by hypoxia in human umbilical vein endothelium. *Circ Res* 2005;97(1):16-24.
- Chen, H., Patterson, N. and Reich, D. Population differentiation as a test for selective sweeps. *Genome Res* 2010;20(3):393-402.

Chen, Y., *et al.* Gene regulatory network stabilized by pervasive weak repressions - microRNA functions revealed by the May-Wigner theory. *National Science Review* 2019.

Collard, C.D., *et al.* Hypoxia-induced expression of complement receptor type 1 (CR1, CD35) in human vascular endothelial cells. *Am J Physiol* 1999;276(2):C450-458.

Deng, L., *et al.* Prioritizing natural selection signals from the deep-sequencing genomic data suggests multi-variant adaptation in Tibetan highlanders. *National Science Review* 2019.

Graham, C.H., Fitzpatrick, T.E. and McCrae, K.R. Hypoxia stimulates urokinase receptor expression through a heme protein-dependent pathway. *Blood* 1998;91(9):3300-3307.

Grossman, S.R., *et al.* A composite of multiple signals distinguishes causal variants in regions of positive selection. *Science* 2010;327(5967):883-886.

Huang, T.S., *et al.* LINC00341 exerts an anti-inflammatory effect on endothelial cells by repressing VCAM1. *Physiol Genomics* 2017;49(7):339-345.

Klemm, S.L., Shipony, Z. and Greenleaf, W.J. Chromatin accessibility and the regulatory epigenome. *Nat Rev Genet* 2019;20(4):207-220.

Kumar, R., Harris-Hooker, S. and Sanford, G. Co-culture of Retinal and Endothelial Cells Results in the Modulation of Genes Critical to Retinal Neovascularization. *Vasc Cell* 2011;3:27.

Losenkova, K., *et al.* Endothelial cells cope with hypoxia-induced depletion of ATP via activation of cellular purine turnover and phosphotransfer networks. *Biochim Biophys Acta Mol Basis Dis* 2018;1864(5 Pt A):1804-1815.

Michiels, C., Arnould, T. and Remacle, J. Endothelial cell responses to hypoxia: initiation of a cascade of cellular interactions. *Biochim Biophys Acta* 2000;1497(1):1-10.

Monteiro, L.J., *et al.* Reduced FOXM1 Expression Limits Trophoblast Migration and Angiogenesis and Is Associated With Preeclampsia. *Reprod Sci* 2019;26(5):580-590.

Nakato, R., *et al.* Comprehensive epigenome characterization reveals diverse transcriptional regulation across human vascular endothelial cells. *Epigenetics Chromatin* 2019;12(1):77.

Namiki, A., *et al.* Hypoxia induces vascular endothelial growth factor in cultured human endothelial cells. *J Biol Chem* 1995;270(52):31189-31195.

Peng, Y., *et al.* Down-Regulation of EPAS1 Transcription and Genetic Adaptation of Tibetans to High-Altitude Hypoxia. *Mol Biol Evol* 2017;34(4):818-830.

Platt, J.L., *et al.* Capture-C reveals preformed chromatin interactions between HIF-binding sites and distant promoters. *EMBO Rep* 2016;17(10):1410-1421.

Pombo, A. and Dillon, N. Three-dimensional genome architecture: players and mechanisms. *Nat Rev Mol Cell Biol* 2015;16(4):245-257.

Punshon, G., *et al.* The long-term stability in gene expression of human endothelial cells permits the production of large numbers of cells suitable for use in regenerative medicine. *Biotechnol Appl Biochem* 2011;58(5):371-375.

Schmedtje, J.F., *et al.* Hypoxia induces cyclooxygenase-2 via the NF-kappaB p65 transcription factor in human vascular endothelial cells. *J Biol Chem* 1997;272(1):601-608.

Shimpuku, H., *et al.* Effect of vitamin E on the degradation of hydrogen peroxide in cultured human umbilical vein endothelial cells. *Life Sci* 2000;68(3):353-359.

Smythies, J.A., *et al.* Inherent DNA-binding specificities of the HIF-1 α and HIF-2 α transcription factors in chromatin. *EMBO Rep* 2019;20(1).

Suzuki, H., *et al.* Paracrine upregulation of VEGF receptor mRNA in endothelial cells by hypoxia-exposed hep G2 cells. *Am J Physiol* 1999;276(1):G92-97.

Therade-Matharan, S., *et al.* Reoxygenation after hypoxia and glucose depletion causes reactive oxygen species production by mitochondria in HUVEC. *Am J Physiol Regul Integr Comp Physiol* 2004;287(5):R1037-1043.

Voellenkle, C., *et al.* Implication of Long noncoding RNAs in the endothelial cell response to hypoxia revealed by RNA-sequencing. *Sci Rep* 2016;6:24141.

Wu, G., *et al.* Hypoxia induces myocyte-dependent COX-2 regulation in endothelial cells: role of VEGF. *Am J Physiol Heart Circ Physiol* 2003;285(6):H2420-2429.

Yoo, S., *et al.* Integrative analysis of DNA methylation and gene expression data identifies EPAS1 as a key regulator of COPD. *PLoS Genet* 2015;11(1):e1004898.

Zhang, P., *et al.* Prenatal hypoxia promotes atherosclerosis via vascular inflammation in the offspring rats. *Atherosclerosis* 2016;245:28-34.

REVIEWERS' COMMENTS:

Reviewer #2 (Remarks to the Author):

The authors have carefully and thoughtfully addressed my previous concerns.

Regarding the remaining concerns of Reviewer no. 1, I went through the authors' response to the reviewer and think they have been very thorough and attentive to the criticisms. I only have two additional comments:

1. They show that the hypoxic DEGs peak and decline over a five day period which is what would be expected based on the literature, and this should be acknowledged.
2. Another rationale for using HUVECs for their study is that it is a cell type that expresses EPAS1 at high levels. Because EPAS1 expression is more tissue restricted than HIF1A (especially in utero), HUVECs are a good bet for this and the other reasons they mention. This should be stated.

Reviewer #3 (Remarks to the Author):

The authors did an excellent job addressing my previous concerns. I support its publication in NC now.

Point-to-point responses to **Reviewers' comments:**

Reviewer #2 (Remarks to the Author):

The authors have carefully and thoughtfully addressed my previous concerns.

Regarding the remaining concerns of Reviewer no. 1, I went through the authors' response to the reviewer and think they have been very thorough and attentive to the criticisms. I only have two additional comments:

1. They show that the hypoxic DEGs peak and decline over a five day period which is what would be expected based on the literature, and this should be acknowledged.

Author's Response: We thank the reviewer's important comment. Indeed, the pattern that DEGs decline over a five-day period was also reported in Peng et al., 2007 [1]. But we can't find any literature systematically discusses the reason and this could be an interesting topic to explore in future. We speculate it might be due to the regulatory feedback loops in hypoxic response derived from the ancient HIF pathway [2], which ensures that oxygen in cell can be maintained after chronic hypoxic induction. We added this statement and cited the references in the main manuscript.

1. Peng, Y. *et al.* Down-Regulation of EPAS1 Transcription and Genetic Adaptation of Tibetans to High-Altitude Hypoxia. *Mol Biol Evol* **34**, 818-830 (2017).
2. Ivan, M. & Kaelin, W.G. The EGLN-HIF O2 Sensing System: Multiple Inputs and Feedbacks. *Mol Cell* **66**, 772-779 (2017).

2. Another rationale for using HUVECs for their study is that it is a cell type that expresses EPAS1 at high levels. Because EPAS1 expression is more tissue restricted than HIF1A (especially in utero), HUVECs are a good bet for this and the other reasons they mention. This should be stated.

Author's Response: We thank the reviewer for the insightful comment on the rationale for using HUVECs. In the revised version, we added this important reason to the Methods ("Choosing relevant cell type to hypoxic response" section) as follow.

"EPAS1 (Endothelial PAS domain protein 1) is a key transcription factor involved in hypoxic induction and the genetic adaptation of Tibetans to high-altitude. Compared with HIF1A, EPAS1 expression is more tissue restricted [3] and highly expressed in endothelial cells, such as HUVEC, which makes it an appropriate cell type in this study."

3. Wiesener, M.S. *et al.* Widespread hypoxia-inducible expression of HIF-2alpha in distinct cell populations of different organs. *FASEB J* **17**, 271-3 (2003).

Reviewer #3 (Remarks to the Author):

The authors did an excellent job addressing my previous concerns. I support its publication in NC now.

We appreciate the reviewer's positive comments.